# Averaging on the Bures-Wasserstein manifold: dimension-free convergence of gradient descent

**Jason M. Altschuler**
MIT
jasonalt@mit.edu

**Sinho Chewi**
MIT
schewi@mit.edu

**Patrik Gerber**
MIT
prgerber@mit.edu

**Austin J. Stromme**
MIT
astromme@mit.edu

## Abstract

We study first-order optimization algorithms for computing the barycenter of Gaussian distributions with respect to the optimal transport metric. Although the objective is geodesically non-convex, Riemannian GD empirically converges rapidly, in fact faster than off-the-shelf methods such as Euclidean GD and SDP solvers. This stands in stark contrast to the best-known theoretical results for Riemannian GD, which depend exponentially on the dimension. In this work, we prove new geodesic convexity results on auxiliary functionals; this provides strong control of the Riemannian GD iterates, ultimately yielding a dimension-free convergence rate. Our techniques also enable the analysis of two related notions of averaging, the entropically-regularized barycenter and the geometric median, providing the first convergence guarantees for Riemannian GD for these problems.

## 1 Introduction

Averaging multiple data sources is among the most classical and fundamental subroutines in data science. However, a modern challenge is that data is often more complicated than points in $\mathbb{R}^d$. In this paper, we study the task of averaging probability distributions on $\mathbb{R}^d$, a setting that commonly arises in machine learning and statistics [CD14; Ho+17; SLD18; Dog+19], computer vision and graphics [Rab+11; Sol+15], probability theory [KS94; RU02], and signal processing [Elv+20]; see also the surveys [PC+19; PZ19] and the references within.

The Wasserstein barycenter [AC11; Rab+11] has emerged as a particularly canonical notion of average. Formally, let $\mathcal{P}_2(\mathbb{R}^d)$ denote the space of probability measures on $\mathbb{R}^d$ with finite second moment, let $P$ be a probability measure over $\mathcal{P}_2(\mathbb{R}^d)$, and let $W_2$ denote the 2-Wasserstein distance (i.e. the standard optimal transport distance). Then, the *Wasserstein barycenter* of $P$ is a solution of

$$\underset{b \in \mathcal{P}_2(\mathbb{R}^d)}{\text{minimize}} \qquad \int W_2^2(b, \cdot) \, \mathrm{d}P \,. \tag{1}$$

A related notion of average is the *entropically-regularized Wasserstein barycenter* of $P$, which is defined to be a solution of

$$\underset{b \in \mathcal{P}_2(\mathbb{R}^d)}{\text{minimize}} \qquad \int W_2^2(b, \cdot) \, \mathrm{d}P + \text{ent}(b) \,, \tag{2}$$

where ent is an entropic penalty which allows for incorporating prior knowledge into the average. Lastly, a third related notion of average with better robustness properties (e.g., with a breakdown point of 50% [FVJ09]) is the *Wasserstein geometric median* of $P$, which is defined to be a solution of

$$\underset{b \in \mathcal{P}_2(\mathbb{R}^d)}{\text{minimize}} \qquad \int W_2(b, \cdot) \, \mathrm{d}P \,. \tag{3}$$

35th Conference on Neural Information Processing Systems (NeurIPS 2021).

Importantly, while these three notions of average can be defined using other metrics in lieu of $W_2$, the Wasserstein distance is critical for many applications since it enables capturing geometric features of the distributions [CD14].

The many applications of Wasserstein barycenters and geometric medians (see e.g., [CE10; Rab+11; CD14; GPC15; RP15; Sol+15; BPC16; SLD18; LLR20]) have inspired significant research into their mathematical and statistical properties since their introduction roughly a decade ago [AC11; Rab+11]. For instance, on the mathematical side it is known that under mild conditions, the barycenter and geometric median exist, are unique, and admit dual formulations related to multimarginal optimal transport problems [CE10; AC11; COO15]. And on the statistical side, [PZ16; AC17; LL17; Big+18; FLF19; ALP20; Le +21; KSS21] provide finite-sample and asymptotic statistical guarantees for estimating the Wasserstein barycenter from samples.

However, computing these objects is challenging because of two fundamental obstacles. The first is that in general, barycenters and geometric medians can be complicated distributions which are much harder to represent (even approximately) than the input distributions. The second is that generically, these problems are computationally hard in high dimensions. For instance, Wasserstein barycenters and geometric medians of discrete distributions are NP-hard to compute (even approximately) in high dimension [AB21b].

**Algorithms for averaging on the Bures-Wasserstein manifold.** Nevertheless, these computational obstacles can be potentially averted in parametric settings. This paper as well as most of the literature [Álv+16; Bac+18; ZP19; Che+20] on parametric settings focuses on the commonly arising setting where $P$ is supported on Gaussian distributions.[1] As noted in [Álv+16], the Gaussian case also encompasses general location-scatter families.

There are two natural families of approaches for designing averaging algorithms in this setting. Both exploit the fact that modulo a simple re-centering of all distributions, the relevant space of probability distributions is isometric to the *Bures-Wasserstein manifold*, i.e. the cone of positive semidefinite matrices equipped with the Bures-Wasserstein metric (background is given in Section 2).

The first approach is simply to recognize the (regularized) Wasserstein barycenter problem as a convex optimization problem over the space of positive semidefinite matrices and apply off-the-shelf methods such as Euclidean GD or semidefinite programming solvers. However, these methods have received little prior attention for good reason: they suffer from severe scalability and parameter-tuning issues (see Section 3.3 for numerics). Briefly, the underlying issue is that these algorithms operate in the standard Euclidean geometry rather than the natural geometry of optimal transport. Moreover, this approach does not apply to the Wasserstein geometric median problem because even in one dimension, it is non-convex in the Euclidean geometry.

A much more effective approach in practice (see Section 3.3 for numerics) is to exploit the geometry of the Bures-Wasserstein manifold via geodesic optimization. Prior work has extensively pursued this direction, investigating the effectiveness of (stochastic) Riemannian GD for computing Wasserstein barycenters, see e.g., [Álv+16; Bac+18; ZP19; Che+20].

**Challenges for geodesic optimization over the Bures-Wasserstein manifold.** Although geodesic optimization is natural for this problem, it comes with several important obstacles: the non-negative curvature of the Bures-Wasserstein manifold necessitates new tools for analysis, and moreover both the barycenter and geometric median problems are *non-convex* in the Bures-Wasserstein geometry. (These two issues are in fact intimately related, see Appendix A.4.) This prevents applying standard results in the geodesic optimization literature (see e.g., [ZS16; Bou20]) since in general it is only possible to prove local convergence guarantees for non-convex problems.

For the Wasserstein barycenter problem, it is possible to interpret Riemannian GD (with step size one) as a fixed-point iteration, and through this lens establish asymptotic convergence [Álv+16; Bac+18; ZP19]. Obtaining non-asymptotic rates of convergence is more challenging because it requires developing quantitative proxies for the standard convexity inequalities needed to analyze GD. The first such result was achieved by [Che+20], showing that Riemannian GD converges to the Wasserstein barycenter at a linear rate. Yet their convergence rate depends exponentially on the

---

[1]In the setting of Gaussians, the Wasserstein barycenter was first studied in the 1990s [OR93; KS94].

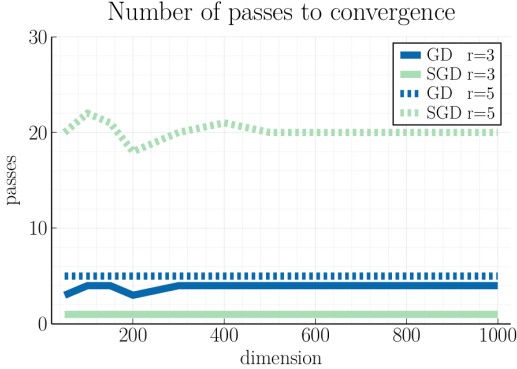

Figure 1: Passes until convergence error $10^{-r}$ to the barycenter, for $r \in \{3, 5\}$. This is *dimension independent* for Riemannian GD and SGD—consistent with our main results. Details in Section 3.

dimension $d$, and also their work does not extend to the Wasserstein geometric median or regularized Wasserstein barycenter.

## 1.1 Contributions

In this paper, we analyze first-order optimization algorithms on the Bures-Wasserstein manifold. We summarize our main results here and overview our techniques in the next section.

**From exponential dimension dependence to dimension-free rates.** In Section 3, we show that for the Wasserstein barycenter problem, Riemannian GD enjoys dimension-free convergence rates (Theorem 2). We make several comments to contextualize this result. First, our result eliminates the exponential dimension dependence of state-of-the-art convergence rates [Che+20], which aligns with the empirical performance of this algorithm (see Figure 1). Second, our result stands in sharp contrast to the setting of discrete distributions in which there are computational complexity barriers to achieving even polynomial dimension dependence [AB21b]. Third, our result closes the gap between computation and statistical estimation for the Bures-Wasserstein barycenter, since dimension-free sample complexity bounds were recently proven in [FLF19].

Moreover, in Theorem 3, we further refine this result by replacing the worst-case assumption of uniform bounds on the matrices' eigenvalues with a significantly weaker average-case assumption.

**Beyond barycenters.** In Sections 4 and 5, we show how our analysis techniques also enable proving fast convergence of Riemannian GD for computing regularized barycenters (Theorem 4) and geometric medians (Theorem 5). To the best of our knowledge, these are the first guarantees for Riemannian GD for notions of averaging on the Bures-Wasserstein manifold beyond the barycenter.

## 1.2 Techniques

Here we briefly sketch the specific technical challenges we face and how we address them to analyze Riemannian GD for the three notions of Bures-Wasserstein average: barycenter, regularized barycenter, and geometric median. Although each analysis necessarily exploits particularities of its own objective, the common structure of our overarching analysis framework may be of interest for studying other geodesically non-convex optimization problems.

**Overcoming non-convexity.** As we discuss in Appendix A.4, there is a close connection between the second-order behavior of these objective functionals and the non-negative curvature of the Bures-Wasserstein manifold. In particular, while non-negative curvature is used to prove smoothness properties for the three functionals, it also leads to them all being geodesically non-convex. To circumvent this issue, we establish gradient domination conditions known as Polyak-Łojasiewicz inequalities [Pol64; Loj63], which intuitively are quantitative proxies for strong convexity in non-convex settings (see e.g., [KNS16; Bol+17]). Proving such inequalities requires synthesizing general optimization principles with specialized arguments based on optimal transport theory. We ultimately

show that these inequalities hold with constants depending on the conditioning of the iterates, i.e., the ratio between the maximum and minimum eigenvalues of the corresponding covariance matrices.

**Overcoming ill-conditioned iterates.**  So long as smoothness and gradient domination inequalities hold at the current iterate, standard optimization results guarantee that the next iterate of GD makes progress. However, the amount of progress degrades if the iterates are poorly conditioned, since then our PL inequality degrades. Thus the second major obstacle is to control the regularity of the iterates. Here, the primary technical tool is shared across the analyses. Informally, it states that if the objective is a sum of functions, each of whose gradients point towards well-conditioned matrices, then the GD iterates remain well-conditioned. Formally, this is captured by the following geodesic convexity result, which may be of independent interest. Below, $\mathbb{S}_{++}^d$ denotes the set of $d \times d$ positive definite matrices. See Appendix A.2 for a review of the relevant geometric concepts, and see Appendix B for the proof, discussion of tightness, and complementary results.

**Theorem 1.** *The functions* $-\sqrt{\lambda_{\min}}, \sqrt{\lambda_{\max}} : \mathbb{S}_{++}^d \to \mathbb{R}$ *are convex along generalized geodesics.*

Using this theorem in conjunction with careful analysis of the objective functions, we establish global convergence guarantees for first-order geodesic optimization.

## 1.3   Other related work

Averages such as barycenters and medians on general curved spaces have become popular due to far-ranging applications in domains such as machine learning, computer vision, analysis, radar signal processing [ABY13], and brain-computer interfaces [YBL16; CBB17]. While their mathematical properties such as existence and uniqueness are fairly well-understood [Afs11], their computation is an active area of research [Wei37; VZ00; Stu03; Yan10; BI13; Bac14; OP15].

For the Wasserstein barycenter problem in particular, there have been many approaches. These approaches vary significantly depending on if the setting is discrete or continuous. In the discrete setting, the problem is NP-hard in high dimension [AB21b]. In low dimension (or more precisely fixed dimension), reasonable approximations can be obtained using fixed-support approximations which reduce the problem to a large linear program [CD14; Ben+15; COO15; Kro+19; Lin+19; Lin+20; Dvi21; Gum+21; Haa+21], and it was recently shown that high-precision (or even exact) solutions can be computed in polynomial time using computational geometry techniques [AB21a].

In the continuous setting, the problem is in general intractable since the optimal barycenter is intractable even to represent, let alone to compute. Nevertheless, in certain settings it has been empirically effective to parameterize and solve using neural networks [CAD20; FTC21; Kor+21], stochastic gradient descent [Li+20], or Riemannian optimization [Álv+16; Bac+18; ZP19; Che+20].

However, for continuous settings, the theory currently lags far behind the empirics. This is true even in the seemingly simple setting of Gaussians, in which case the barycenter has a concise representation since it is also Gaussian. Riemmanian GD for this problem was first proposed and demonstrated to be empirically effective in [Álv+16], where it was introduced as a fixed-point algorithm. Asymptotic convergence was proved in [Álv+16], and then extended to non-population and stochastic settings in [Bac+18; ZP19]. Non-asymptotic convergence rates were first shown in [Che+20], but there is a large gap between these theoretical rates and what is observed in practice. It particular, previous rates depend exponentially on the dimension. The present paper improves this to dimension-free.

For the other two problems we study, Bures-Wasserstein geometric medians and entropically-regularized barycenters, no convergence guarantees were previously known for the natural Riemannian GD algorithm.

Our work is in the midst of a flurry of exciting recent developments about entropically regularized barycenters. Several recent works have, simultaneously with each other, extensively studied these objects in the particular setting of Gaussians, leading to the establishment of fundamental results such as the fact that the regularized barycenter of Gaussians is itself Gaussian [BL20; Jan+20; MGM21]. Another recent and related line of work has established fundamental mathematical and statistical results for entropically regularized barycenters in the setting of general distributions, although with slightly different penalties than the KL divergence studied here, typically the differential entropy $\int b \ln b$ [Kro18; BCP19; CEK21] and sometimes even broader classes of regularizations [BCP19].

### 1.4 Organization

Section 2 briefly recalls relevant preliminaries. We analyze Riemannian GD for computing Bures-Wasserstein barycenters, regularized barycenters, and geometric medians in Sections 3, 4, and 5, respectively. We conclude in Section 6. We provide proofs as well as additional background and numerical results in the Appendix. Code reproducing all experiments in this paper can be found at https://github.com/PatrikGerber/Bures-Barycenters.

## 2 Preliminaries

Given probability measures $\mu$ and $\nu$ on $\mathbb{R}^d$ with finite second moment, the 2-Wasserstein distance between $\mu$ and $\nu$ is defined as

$$W_2^2(\mu, \nu) := \inf_{\pi \in \Pi(\mu, \nu)} \int \|x - y\|^2 \, \mathrm{d}\pi(x, y) \,, \tag{4}$$

where $\Pi(\mu, \nu)$ denotes the set of couplings of $\mu$ and $\nu$, i.e., the probability measures on $\mathbb{R}^d \times \mathbb{R}^d$ whose marginals are respectively $\mu$ and $\nu$. If $\mu$ and $\nu$ admit densities with respect to the Lebesgue measure on $\mathbb{R}^d$, then the infimum is attained, and the optimal coupling is supported on the graph of a map, i.e., there exists a map $T : \mathbb{R}^d \to \mathbb{R}^d$ such that for $\pi$-a.e. $(x, y) \in \mathbb{R}^d \times \mathbb{R}^d$, it holds that $y = T(x)$. The map $T$ is called the *optimal transport map* from $\mu$ to $\nu$. We refer readers to [Vil03; San15] for an introduction to optimal transport, and to [Car92] for background on Riemannian geometry. The Riemannian structure of optimal transport was introduced in the seminal work [Ott01]—detailed treatments are in [AGS08; Vil09]; for completeness we also provide a quick overview in Appendix A.

In this paper, we mainly work with centered Gaussians, which can be identified with their covariance matrices. (Extensions to the non-centered case are also discussed below.) We abuse notation via this identification: given $\Sigma, \Sigma' \in \mathbb{S}_{++}^d$, we write $W_2(\Sigma, \Sigma')$ for the 2-Wasserstein distance between centered Gaussians with covariance matrices $\Sigma$, $\Sigma'$ respectively. Here, $\mathbb{S}^d$ denotes the space of symmetric $d \times d$ matrices, and $\mathbb{S}_{++}^d$ denotes the open subset of $\mathbb{S}^d$ consisting of positive definite matrices. Throughout, all Gaussians are non-degenerate; that is, their covariances are non-singular.

The Wasserstein distance has a closed-form expression for Gaussians:

$$W_2^2(\Sigma, \Sigma') = \mathrm{tr}\left[\Sigma + \Sigma' - 2\left(\Sigma^{1/2}\Sigma'\Sigma^{1/2}\right)^{1/2}\right]. \tag{5}$$

Also, the optimal transport map from $\Sigma$ to $\Sigma'$ is the symmetric matrix

$$T_{\Sigma \to \Sigma'} = \Sigma^{-1/2}\left(\Sigma^{1/2}\Sigma'\Sigma^{1/2}\right)^{1/2}\Sigma^{-1/2} = \mathbf{GM}(\Sigma^{-1}, \Sigma'). \tag{6}$$

Above, $\mathbf{GM}(A, B) := A^{1/2}\left(A^{-1/2}BA^{-1/2}\right)^{1/2}A^{1/2}$ denotes the matrix geometric mean between two positive semidefinite matrices [Bha07, Ch. 4]. The Wasserstein distance on $\mathbb{S}_{++}^d$ in fact arises from a Riemannian metric, which was first introduced by Bures in [Bur69]. Hence, the Riemannian manifold $\mathbb{S}_{++}^d$ endowed with this Wasserstein distance is referred to as the *Bures-Wasserstein space*. The geometry of this space is studied in detail in [Mod17; BJL19]. For completeness, we provide additional background on the Bures-Wasserstein manifold in Appendix A.

## 3 Barycenters

In this section, we consider the Bures-Wasserstein barycenter

$$\Sigma^\star \in \operatorname*{arg\,min}_{\Sigma \in \mathbb{S}_{++}^d} \int W_2^2(\Sigma, \cdot) \, \mathrm{d}P \,.$$

We refer to the introduction for a discussion of the past work on the Bures-Wasserstein barycenter. We also remark that the case when $P$ is supported on possibly non-centered Gaussians is easily reduced to the centered case; see the discussion in [Che+20, §4].

## 3.1 Algorithms

We consider both Riemannian gradient descent (GD) and Riemannian stochastic gradient descent (SGD) algorithms for computing the Bures-Wasserstein barycenter, which are given as Algorithm 1 and Algorithm 2 respectively. GD is useful for computing high-precision solutions due to its linear convergence (Theorem 2), and SGD is useful for large-scale or online settings because of its cheaper updates. We refer to [ZP19; Che+20] for the derivation of the updates. Here, $\Sigma_0$ is the initialization, which can be taken to be any matrix in the support of $P$. For SGD, we also require a sequence $(\eta_t)_{t=1}^T$ of step sizes and a sequence $(K_t)_{t=1}^T$ of i.i.d. samples from $P$. Note that whereas SGD requires choosing step sizes, GD simply uses step size 1, as justified in [ZP19].

| **Algorithm 1** GD for Barycenters |
| --- |
| 1: **procedure** BARY-GD$(\Sigma_0, P, T)$ |
| 2:      **for** $t = 1, \ldots, T$ **do** |
| 3:          $S_t \leftarrow \int \mathrm{GM}(\Sigma_{t-1}^{-1}, \Sigma) \, \mathrm{d}P(\Sigma)$ |
| 4:          $\Sigma_t \leftarrow S_t \Sigma_{t-1} S_t$ |
| 5:      **return** $\Sigma_T$ |

| **Algorithm 2** SGD for Barycenters |
| --- |
| 1: **procedure** BARY-SGD$(\Sigma_0, (\eta_t)_{t=1}^T, (K_t)_{t=1}^T)$ |
| 2:      **for** $t = 1, \ldots, T$ **do** |
| 3:          $\hat{S}_t \leftarrow (1 - \eta_t) I_d + \eta_t \, \mathrm{GM}(\Sigma_{t-1}^{-1}, K_t)$ |
| 4:          $\Sigma_t \leftarrow \hat{S}_t \Sigma_{t-1} \hat{S}_t$ |
| 5:      **return** $\Sigma_T$ |

## 3.2 Convergence guarantees

Denote the barycenter functional by $F(\Sigma) := \frac{1}{2} \int W_2^2(\Sigma, \cdot) \, \mathrm{d}P$, and denote the *variance* of $P$ by $\mathrm{var}\, P := 2F(\Sigma^\star)$. We assume that $P$ is supported on matrices whose eigenvalues lie in the range $[\lambda_{\min}, \lambda_{\max}]$, and we let $\kappa := \lambda_{\max}/\lambda_{\min}$ denote the condition number. Whereas the previous state-of-the-art convergence analysis for Algorithms 1 and 2 in [Che+20] suffered a dependence of $\kappa^d$, we show that the rates of convergence are in fact independent of the dimension $d$.

**Theorem 2.** *Assume that $P$ is supported on covariance matrices whose eigenvalues lie in the range $[\lambda_{\min}, \lambda_{\max}]$, $0 < \lambda_{\min} \leq \lambda_{\max} < \infty$. Let $\kappa := \lambda_{\max}/\lambda_{\min}$ denote the condition number. Assume that we initialize at $\Sigma_0 \in \mathrm{supp}\, P$.*

1. *(GD) Let $\Sigma_T^{\mathrm{GD}}$ denote the $T$-th iterate of GD (Algorithm 1). Then,*

$$\frac{1}{2\sqrt{\kappa}} W_2^2(\Sigma_T^{\mathrm{GD}}, \Sigma^\star) \leq F(\Sigma_T^{\mathrm{GD}}) - F(\Sigma^\star) \leq \exp\left(-\frac{T}{4\kappa^{3/2}}\right) \{F(\Sigma_0) - F(\Sigma^\star)\}.$$

2. *(SGD) Let $\Sigma_T^{\mathrm{SGD}}$ denote the $T$-th iterate of SGD (Algorithm 2), where $(K_t)_{t=1}^T$ are i.i.d. from $P$. Then, with an appropriate choice of step sizes[2],*

$$\frac{1}{2\sqrt{\kappa}} \mathbb{E}\, W_2^2(\Sigma_T^{\mathrm{SGD}}, \Sigma^\star) \leq \mathbb{E}\, F(\Sigma_T^{\mathrm{SGD}}) - F(\Sigma^\star) \leq \frac{48\kappa^3 \, \mathrm{var}\, P}{T}.$$

In fact, using our new geodesic convexity results we can also relax the conditioning assumption from requiring all matrices be *uniformly* well-conditioned, to being *individually* well-conditioned. This is a significant improvement when the eigenvalue ranges differ significantly between matrices.

**Theorem 3.** *Let $\kappa^\star := \sup_{\Sigma \in \mathrm{supp}(P)} \lambda_{\max}(\Sigma)/\lambda_{\min}(\Sigma)$. The conclusions of Theorem 2 hold when replacing $\kappa$ with $\kappa^\star$ everywhere.*

Actually, we deduce this from an even stronger statement: in Theorem 2, $\kappa$ can be replaced everywhere by an *average-case* notion of conditioning, namely $\overline{\kappa} := \overline{\lambda_{\max}}/\overline{\lambda_{\min}}$ where $\overline{\lambda_{\min}}^{1/2} := \int \lambda_{\min}(\Sigma)^{1/2} \, \mathrm{d}P(\Sigma)$ and $\overline{\lambda_{\max}}^{1/2} := \int \lambda_{\max}(\Sigma)^{1/2} \, \mathrm{d}P(\Sigma)$.

We give the proofs of these results in Appendix C.1.

---

[2]Namely, $\eta_t = \frac{1}{4\kappa^{3/2}} \left(1 - \sqrt{1 - \frac{16\kappa^3 (2(t+t_0)+1)}{(t+t_0+1)^2}}\right)$ suffices, where $t_0 = 32\kappa^3 - 1$.

### 3.3 Numerical experiments

There are two natural competitors of Riemannian GD when minimizing the barycenter functional: ($i$) solving an SDP (see Appendix C.3 for the SDP reformulation), and ($ii$) Euclidean GD (see Appendix C.2 for a description and analysis of Euclidean GD).

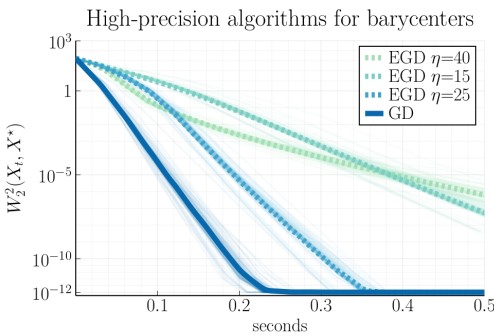

Figure 2: Riemannian vs. Euclidean GD.

Figure 3: Riemannian vs. Euclidean SGD.

In Figure 2 we compare Riemannian and Euclidean GD on a dataset consisting of $n = 50$ covariance matrices of dimension $d = 50$, each with condition number $\kappa = 1000$. Their eigenspaces are independent Haar distributed, and their eigenvalues are equally spaced in the interval $[\lambda_{\min}, \lambda_{\max}] = [0.03, 30]$. Qualitatively similar results are observed for other inputs; see Appendix F. We run 50 experiments and plot the average accuracy cut off at $10^{-12}$; $X^\star$ denotes the best iterate. We omit SDP solvers from the plot because their runtime is orders of magnitude slower: using the Splitting Cone Solver (SCS) [ODo+16; ODo+19], the problem takes $\sim$15 seconds to solve, and MOSEK [MOS21] is even slower. For completeness, we also compare GD to the Riemannian Frank-Wolfe algorithm [WS17] in Appendix F, and conclude that GD is superior. We observe that Euclidean GD's convergence rate is quite sensitive to its step size, which depends heavily on the conditioning of the problem. Riemannian GD was the clear winner in our experiments, as its step size requires no tuning and it always performed no worse (in fact, often significantly better) than Euclidean GD.

In Figure 3 we observe that Riemannian SGD typically outperforms Euclidean SGD, sometimes substantially. We average $300 \times 300$ covariance matrices drawn from a distribution whose barycenter is known to be the identity, see Appendix F for details. As Figure 3 shows, in practice it can be helpful to tune the step sizes beyond the guidance given by our worst-case theoretical guarantees. In our experiments, Riemannian SGD was competitive on a wide range of problems with $\eta = 1/t$.

We comment on Figure 1, which illustrates the dimension independence of the two Riemannian algorithms, a main result of this paper. It plots the number of passes until convergence $W_2^2(X_t, X^\star) \leq 10^{-r} \operatorname{var} P$ to the barycenter $X^\star$, for $r \in \{3, 5\}$. To compare algorithms on equal footing, the $y$-axis measures "passes" over the $n = 50$ matrices: a pass constitutes one GD iteration, or $n$ SGD iterations. The input is generated as in Figure 2. Observe also the tradeoff between GD and SGD: SGD converges rapidly to low-precision solutions, but takes longer to converge to high-precision solutions.

## 4 Entropically-regularized barycenters

In this section, we consider the entropically-regularized barycenter $b_{\mathrm{reg}}^\star$ which minimizes

$$F_\gamma(b) := \frac{1}{2} \int W_2^2(b, \cdot) \, \mathrm{d}P + \gamma \operatorname{KL}\!\left(b \,\big\|\, \mathcal{N}(0, I_d)\right),$$

where KL denotes the Kullback-Leibler divergence, and $\gamma > 0$ is a given regularization parameter. It suffices to consider the case when all measures are centered, see Remark 6. The following proposition justifies considering this problem on the Bures-Wasserstein space; proof in Appendix D.3.

**Proposition 1.** *Suppose $P$ is supported on centered Gaussians whose covariance matrices have eigenvalues lying in the range $[1/\sqrt{\kappa}, \sqrt{\kappa}]$, for some $\kappa \geq 1$. Then there exists a unique minimizer $b_{\mathrm{reg}}^\star$ of $F_\gamma$ over $\mathcal{P}_2(\mathbb{R}^d)$, and this minimizer is a centered Gaussian distribution whose covariance matrix $\Sigma^\star$ also has eigenvalues in the range $[1/\sqrt{\kappa}, \sqrt{\kappa}]$.*

As described in the introduction, prior work on the Wasserstein barycenter typically focuses on a slightly different entropic penalty, the differential entropy $\int b \ln b$. Note that the differential entropy penalty encourages $b$ to be diffuse over all of $\mathbb{R}^d$ (the minimizer blows up as $\gamma \to \infty$). Here, we focus on a KL divergence penalty which has the advantage of interpolating between two well-studied problems: the Wasserstein barycenter problem ($\gamma = 0$) and minimization of the KL divergence ($\gamma = \infty$). We take the standard Gaussian as a canonical choice of reference distribution, and note that our method of analysis can be extended to other reference measures at the cost of significant additional technical complexity. We thus choose to exclusively focus on the standard Gaussian case.

### 4.1 Algorithm

Algorithm 3 is Riemannian GD for minimizing $F_\gamma$. A derivation of the update rule is in Appendix D.

---
**Algorithm 3** GD for Regularized Barycenters
---
1: **procedure** RBARY-GD($\Sigma_0, P, T, \gamma, \eta$)
2:      **for** $t = 1, \ldots, T$ **do**
3:          $S_t \leftarrow \eta \int \mathrm{GM}(\Sigma_{t-1}^{-1}, \Sigma)\, \mathrm{d}P(\Sigma) + \eta\gamma\Sigma_{t-1}^{-1} + (1 - \eta\,(1+\gamma))I_d$
4:          $\Sigma_t \leftarrow S_t\Sigma_{t-1}S_t$
5:      **return** $\Sigma_T$
---

### 4.2 Convergence guarantees

We provide two convergence guarantees for Algorithm 3. The first holds for all choices of the regularization parameter $\gamma$. However, this rate deteriorates with larger $\gamma$, and intuitively the optimization problem should become somewhat easier with larger regularization; hence, we prove a second rate of convergence to capture this behavior. We emphasize that as in §3, our convergence rates are dimension-independent. The proof of each rate appears in Appendix D.

**Theorem 4.** *Fix $\gamma > 0$ and suppose that $P$ is supported on covariance matrices with eigenvalues in $[1/\sqrt{\kappa}, \sqrt{\kappa}]$. If Algorithm 3 is initialized at a point in* supp $P$ *and run with step size $\eta = 1/(1 + 2\gamma\sqrt{\kappa})$, then the following hold.*

1. *For any choice of regularization parameter $\gamma > 0$ and any $T \geq 1$,*

$$F_\gamma(\Sigma_T) - F_\gamma(\Sigma^\star) \leq \exp\left(-\frac{T}{\kappa^4\,(1 + 2\gamma\sqrt{\kappa})}\right)\{F_\gamma(\Sigma_0) - F_\gamma(\Sigma^\star)\}.$$

2. *If $\gamma \geq 14\kappa^4$ is sufficiently large, then the following improved rate holds: for $T \geq 1$,*

$$F_\gamma(\Sigma_T) - F_\gamma(\Sigma^\star) \leq \exp\left(-\frac{T}{6\sqrt{\kappa}}\right)\{F_\gamma(\Sigma_0) - F_\gamma(\Sigma^\star)\}.$$

For brevity, we omit guarantees in terms of the distance $W_2(\Sigma_t, \Sigma^\star)$ to the minimizer.

### 4.3 Numerical experiments

In Figure 4, we investigate the use of the regularization term $\gamma \mathrm{KL}(\cdot \,\|\, \mathcal{N}(0, I_d))$ to encode a prior belief of isotropy. In Figure 4, we generate $n = 100$ i.i.d. $20 \times 20$ covariance matrices from a distribution whose barycenter is the identity (see Appendix F). Then, for $\rho \in [0, 10]$ we compute the barycenter of a perturbed dataset obtained by adding $\rho e_1 e_1^\mathsf{T}$ to each matrix for different choices of $\gamma$. We see that for $\gamma = 0$ the barycenter quickly departs from isotropy, while for larger $\gamma$ the regularization yields averages which are more consistent with our prior belief.

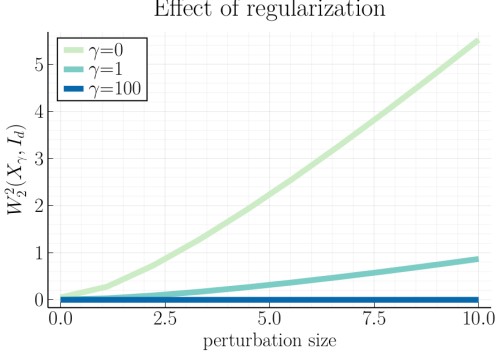

Figure 4: Effect of regularization for varying $\gamma$.

# 5 Geometric medians

In this section, we consider the Wasserstein geometric median

$$b^\star_{\text{median}} \in \underset{b \in \mathcal{P}_2(\mathbb{R}^d)}{\arg\min} \int W_2(b, \cdot) \, dP \,. \tag{7}$$

See the introduction for a discussion of the literature on this problem. Observe that, in contrast to the barycenter (1), here we are minimizing the average *unsquared* Wasserstein distance.

The following basic result justifies the consideration of the geometric median problem on the Bures-Wasserstein space. It is proved in Appendix E.1.

**Proposition 2.** *Suppose that $P$ is supported on centered non-degenerate Gaussians whose covariance matrices have eigenvalues lying in the range $[\lambda_{\min}, \lambda_{\max}]$, where $0 \le \lambda_{\min} \le \lambda_{\max} < \infty$. Then, there exists a solution to* (7) *which is also a centered non-degenerate Gaussian distribution; moreover, its covariance matrix $\Sigma^\star_{\text{median}}$ can be taken to have eigenvalues in $[\lambda_{\min}, \lambda_{\max}]$.*

*Remark* 1. Suppose now that $P$ is supported on non-degenerate Gaussian distributions which are not necessarily centered. Then, the proof of Proposition 2 applies with minor modifications to show that the minimizer of the median functional is still attained at a Gaussian distribution. However, unlike the barycenter and entropically regularized barycenter, it is not the case that the mean of the Wasserstein geometric median is the Euclidean geometric median of the means, thus it is not as straightforward to reduce to the centered case for this problem. Nevertheless, in Appendix E.2, we describe a reduction which allows the algorithm described in the next section to be applied in a black box manner to the non-centered case, with corresponding convergence guarantees.

## 5.1 Algorithm

Since the Wasserstein distance $W_2(\Sigma, \cdot)$ is neither geodesically convex nor geodesically smooth, nor Euclidean convex nor Euclidean smooth (see Remark 7), it poses challenges for optimization. We therefore smooth the objective before optimization. Given a desired target accuracy $\varepsilon > 0$, let

$$W_{2,\varepsilon} := \sqrt{W_2^2 + \varepsilon^2} \,, \qquad F_\varepsilon(b) := \int W_{2,\varepsilon}(b, \cdot) \, dP \,.$$

Algorithm 4 provides pseudocode for running Riemannian GD on the smoothed functional $F_\varepsilon$. See Appendix E for a derivation of the update rule.

---

**Algorithm 4** Smoothed GD for Median

---

1: **procedure** MEDIAN-GD($\Sigma_0, P, T, \varepsilon$)
2:     **for** $t = 1, \ldots, T$ **do**
3:         $S_t \leftarrow I_d + \varepsilon \int \{\text{GM}(\Sigma_{t-1}^{-1}, \Sigma) - I_d\} \, W_{2,\varepsilon}(\Sigma_{t-1}, \Sigma)^{-1} \, dP(\Sigma)$
4:         $\Sigma_t \leftarrow S_t \Sigma_{t-1} S_t$
5:     **return** $\Sigma_T$

---

## 5.2 Convergence guarantees

We show that Algorithm 4 finds an $\mathcal{O}(\varepsilon)$-approximate minimizer for the geometric median functional in $\mathcal{O}(\kappa/\varepsilon^4)$ iterations. We emphasize that as in our other results, this convergence is dimension-independent. Below, let $F := F_0$ denote the unregularized functional. Note that since $F$ typically does not have a unique minimizer, we only guarantee a small suboptimality. The proof is in Appendix E.1.

**Theorem 5.** *Assume that $P$ is supported on covariance matrices whose eigenvalues lie in $[\lambda_{\min}, \lambda_{\max}]$, $0 < \lambda_{\min} \le \lambda_{\max} < \infty$. Let $\kappa := \lambda_{\max}/\lambda_{\min}$ denote the condition number, and let $0 < \varepsilon < 1$ denote a target accuracy. Assume that we initialize Algorithm 4 at $\Sigma_0 \in \text{supp } P$. Then, Algorithm 4 outputs $\Sigma_T$ satisfying $F(\Sigma_T) - F(\Sigma^\star_{\text{median}}) \le 3\varepsilon$ if*

$$T \ge \frac{32\kappa \, F_\varepsilon(\Sigma_0)^4}{\varepsilon^4} \,.$$

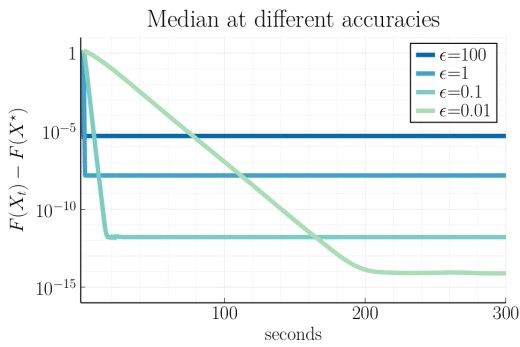
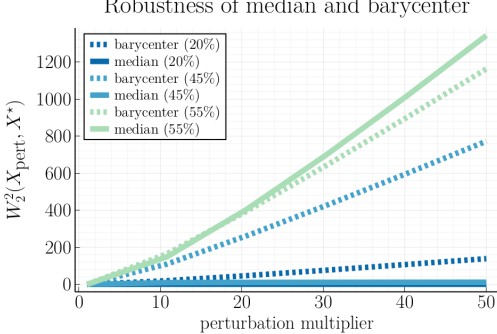

Figure 5: Evolution of median objective for varying $\varepsilon$. $X^\star$ denotes the best iterate.

Figure 6: Robustness of the Wasserstein median.

### 5.3 Numerical experiments

In Figure 5 we plot the suboptimality gap for the unregularized objective $F = \int W_2(\cdot, \Sigma) \, dP$ as we optimize $F_\varepsilon$ using Algorithm 4 for various $\varepsilon$. The regularization parameter $\varepsilon$ has a natural trade-off: smaller $\varepsilon$ results in better approximation to the (unregularized) geometric median, but slower convergence. The covariance matrices are generated as in Figure 2, with $n = d = 30$ and $[\lambda_{\min}, \lambda_{\max}] = [0.01, 10]$. The promising empirics in Figure 5 suggest that our algorithm performs even better in practice than our worst-case theoretical results guarantee: few iterations may suffice for convergence, and also moderate regularization may suffice for high-precision approximations.

In Figure 6 we illustrate the robustness of the Wasserstein geometric median up to its breakdown point of $50\%$ [FVJ09]. We take random input matrices as above, with $n = d = 20$ and $[\lambda_{\min}, \lambda_{\max}] = [1, 10]$, and compute their barycenter and approximate median ($\varepsilon = 1$). We then perturb a fraction ($20\%$, $45\%$, and $55\%$ for our figure) of the matrices by multiplying them by a constant greater than 1. The $x$-axis of the plot shows the size of the perturbation while the $y$-axis gives the distance of the original barycenter and median to the barycenter and median of this new, perturbed dataset.

We also implemented Euclidean GD for this geometric median problem; plots are omitted for brevity since the results are similar to those for the barycenter (c.f. Section 3.3) in that Euclidean GD depends much more heavily on parameter tuning. Note also that Euclidean GD does not come with convergence guarantees for this problem since it is non-convex in the Euclidean geometry.

## 6 Discussion

In this paper we revisited the problem of computing Bures-Wasserstein barycenters and explained the empirical efficacy of Riemannian (S)GD by proving convergence rates that improve from exponential dimension dependence to dimension-free. An attractive feature of our analysis framework was that our tools were sufficiently general to prove similar dimension-free guarantees for related problems of interest, namely Bures-Wasserstein geometric medians and entropically-regularized barycenters.

Our results suggest several interesting directions for future research. The focus of this paper was dimension-dependence, and while we also improved the dependence on other parameters along the way, it is unclear if these other dependencies are optimal. Can the dependence on $\kappa$ be improved via stronger PL inequalities? Is the dependence on $\varepsilon$ improvable via alternate methods of smoothing in the case of geometric medians, or via fixed-point acceleration schemes such as Anderson acceleration in the case of barycenters?

More broadly, conventional wisdom from the now-established field of geodesic optimization tells us that whenever possible, one should recast a non-convex optimization problem as a convex one by changing the geometry. However, as demonstrated empirically in Section 3, for computing Bures-Wasserstein barycenters, it is significantly better to run GD in the non-convex geometry of optimal transport than in the convex geometry of Euclidean space. A full understanding of why and when non-convex geometry can be helpful in general optimization problems is an intriguing direction with potentially significant implications for both the theory and practice of non-convex optimization.

## Acknowledgments and Disclosure of Funding

We are grateful to Victor-Emmanuel Brunel, Tyler Maunu, and Philippe Rigollet for stimulating conversations, to Pablo Parrilo for pointing out that Bures-Wasserstein barycenters have an SDP formulation (Appendix C.3), and to the anonymous reviewers for their thoughtful comments.

JA was supported by NSF Graduate Research Fellowship 1122374 and a TwoSigma PhD fellowship. SC and AS were supported by the Department of Defense (DoD) through the National Defense Science & Engineering Graduate Fellowship (NDSEG) Program.

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
