# Appendix

## Table of Contents

## A  Background on the Bures-Wasserstein manifold

In this section, we collect relevant background about Bures-Wasserstein geometry to make the paper more self-contained.

### A.1  Geometry

We begin by describing the geometry of optimal transport, and then explain how to specialize the general concepts to the Bures-Wasserstein manifold. The books [AGS08; Vil09] are definitive references for the Riemannian structure of optimal transport. We attempt to convey the main relevant ideas, and in doing so do not attempt to be fully rigorous here.

Let $\mathcal{P}_{2,\mathrm{ac}}(\mathbb{R}^d)$ denote the space of all probability measures on $\mathbb{R}^d$ which are absolutely continuous (i.e. admit a density w.r.t. the Lebesgue measure) and which have a finite second moment. When equipped with the 2-Wasserstein distance $W_2$, it becomes a metric space. In fact, more is true: $(\mathcal{P}_{2,\mathrm{ac}}(\mathbb{R}^d), W_2)$ admits a formal Riemannian structure which we now describe. Given $\mu_0, \mu_1 \in \mathcal{P}_{2,\mathrm{ac}}(\mathbb{R}^d)$, let $T$ denote the optimal transport map from $\mu_0$ to $\mu_1$; thus, $T : \mathbb{R}^d \to \mathbb{R}^d$ is a map satisfying $T_\#\mu_0 = \mu_1$.

Here, $\#$ denotes the pushforward operation, i.e. if $X \sim \mu_0$, then $T(X) \sim \mu_1$. The constant-speed geodesic $(\mu_t)_{t \in [0,1]}$ joining $\mu_0$ to $\mu_1$ is described via

$$\mu_t = [(1-t)\,\mathrm{id} + tT]_\# \mu_0\,, \qquad t \in [0,1]\,.$$

This geodesic has the following interpretation: draw a "particle" $X_0 \sim \mu_0$, and move $X_0$ to $T(X_0)$ with constant speed for one unit of time along the Euclidean geodesic (i.e. straight line) joining these endpoints; thus, at time $t$, the particle is at position $X_t = (1-t)X_0 + tT(X_0)$. Then, $\mu_t$ is simply the law of $X_t$.

We take the tangent vector of the geodesic $(\mu_t)_{t \in [0,1]}$ at time 0 to be the mapping $T - \mathrm{id}$; note that in the particle view, $T(X_0) - X_0$ represents the velocity of the particle at time 0. The tangent space $T_{\mu_0}\mathcal{P}_{2,\mathrm{ac}}(\mathbb{R}^d)$ to $\mathcal{P}_{2,\mathrm{ac}}(\mathbb{R}^d)$ at $\mu_0$ is then defined to consist of all possible tangent vectors to geodesics emanating from $\mu_0$. Actually, in order to make $T_{\mu_0}\mathcal{P}_{2,\mathrm{ac}}(\mathbb{R}^d)$ formally into a (closed subset of a) Hilbert space, the definition is modified to read ([AGS08, Theorem 8.5.1])

$$T_{\mu_0}\mathcal{P}_{2,\mathrm{ac}}(\mathbb{R}^d) := \overline{\{\lambda\,(T_{\mu_0 \to \nu} - \mathrm{id}) : \lambda > 0,\ \nu \in \mathcal{P}_{2,\mathrm{ac}}(\mathbb{R}^d)\}}^{L^2(\mu_0)}\,, \tag{8}$$

where the overline denotes the $L^2(\mu_0)$ closure; we equip this tangent space with the $L^2(\mu_0)$ norm. Thus, for instance, we have $W_2^2(\mu_0, \mu_1) = \mathbb{E}[\|X_0 - T_{\mu_0 \to \mu_1}(X_0)\|^2] = \|T_{\mu_0 \to \mu_1} - \mathrm{id}\|_{L^2(\mu_0)}^2$, which says that the squared norm of the tangent vector of the geodesic $(\mu_t)_{t \in [0,1]}$ equals the squared Wasserstein distance. We may write $\|\cdot\|_{\mu_0}$ as a shorthand for $\|\cdot\|_{L^2(\mu_0)}$.

The Riemannian exponential map $\exp_\mu$ is the mapping $T_\mu \mathcal{P}_{2,\mathrm{ac}}(\mathbb{R}^d) \to \mathcal{P}_{2,\mathrm{ac}}(\mathbb{R}^d)$ which maps a tangent vector $v$ to the constant-speed geodesic emanating from $\mu$ with velocity $v$, evaluated at time 1.[3] From our description above, we see that $\exp_\mu v = (\mathrm{id} + v)_\# \mu$, since the tangent vector joining $\mu$ to $T_\# \mu$ is $v = T - \mathrm{id}$ (when $T$ is an optimal transport map). It is also convenient to define the Riemannian logarithmic map $\log_\mu : \mathcal{P}_{2,\mathrm{ac}}(\mathbb{R}^d) \to T_\mu \mathcal{P}_{2,\mathrm{ac}}(\mathbb{R}^d)$ to be the inverse of the exponential map $\exp_\mu$; in our context, $\log_\mu \nu = T_{\mu \to \nu} - \mathrm{id}$.

In Riemannian geometry, it is common to localize the argument around a measure $\mu$, which loosely means replacing a measure $\nu$ with its image $\log_\mu \nu$ in the tangent space at $\mu$. This is convenient because the tangent space at $\mu$ is embedded in the Hilbert space $L^2(\mu)$, and we can leverage Hilbert space arguments (e.g. computing inner products). In order to do this one must quantify the distortion introduced by the map $\log_\mu$, which is morally related to the curvature of the manifold.

We now specialize the above concepts to the Bures-Wasserstein manifold, in which non-degenerate centered Gaussians are identified with their covariance matrices; thus, the Bures-Wasserstein manifold is the space $\mathbb{S}_{++}^d$ of positive-definite symmetric matrices equipped with a certain Riemannian metric.

The optimal transport problem between Gaussians is discussed in many places, e.g. [BJL19]. Given two covariance matrices $\Sigma, \Sigma' \in \mathbb{S}_{++}^d$, the optimal transport map between the corresponding centered Gaussians is the linear map $\mathbb{R}^d \to \mathbb{R}^d$ given by

$$T_{\Sigma \to \Sigma'} = \Sigma^{-1/2}\left(\Sigma^{1/2}\Sigma'\Sigma^{1/2}\right)^{1/2}\Sigma^{-1/2}\,.$$

Note that this is a symmetric matrix. Since $AX \sim \mathcal{N}(0, A\Sigma A^\mathsf{T})$ for $X \sim \mathcal{N}(0, \Sigma)$, the fact that $T_{\Sigma \to \Sigma'} X \sim \mathcal{N}(0, \Sigma')$ reduces to the matrix identity $T_{\Sigma \to \Sigma'} \Sigma T_{\Sigma \to \Sigma'} = \Sigma'$, which can be verified by hand. The above formula yields

$$\begin{aligned} W_2^2(\Sigma, \Sigma') &= \mathbb{E}[\|X - T_{\Sigma \to \Sigma'} X\|^2] = \mathbb{E}[\|X\|^2 + \|T_{\Sigma \to \Sigma'} X\|^2 - 2\,\langle X, T_{\Sigma \to \Sigma'} X\rangle] \\ &= \mathrm{tr}(\Sigma + \Sigma' - 2\Sigma T_{\Sigma \to \Sigma'})\,. \end{aligned} \tag{9}$$

From the general description of Wasserstein geodesics, the constant-speed geodesic $(\Sigma_t)_{t \in [0,1]}$ joining $\Sigma$ to $\Sigma'$ is given by

$$\Sigma_t = \left((1-t)I_d + tT_{\Sigma \to \Sigma'}\right)\Sigma\left((1-t)I_d + tT_{\Sigma \to \Sigma'}\right)\,, \qquad t \in [0,1]\,. \tag{10}$$

---

[3]Generally, in Riemannian geometry, the exponential map is not defined on the entire tangent space but rather a subset of it; this is also the case for Wasserstein space.

The tangent space $T_\Sigma \mathbb{S}_{++}^d$ is identified with the space $\mathbb{S}^d$ of symmetric $d \times d$ matrices. Given $S \in T_\Sigma \mathbb{S}_{++}^d$, the tangent space norm of $S$ is given by $\|S\|_{L^2(\mathcal{N}(0,\Sigma))} = \sqrt{\mathbb{E}[\|SX\|^2]} = \sqrt{\langle S^2, \Sigma \rangle}$, which we simply denote as $\|S\|_\Sigma$. More generally, given matrices $A$, $B$, we write $\langle A, B \rangle_\Sigma := \mathrm{tr}(A^\mathsf{T} \Sigma B)$. The exponential map[4] is $\exp_\Sigma S = (I_d + S)\,\Sigma\,(I_d + S)$, so that $\exp_\Sigma(T_{\Sigma \to \Sigma'} - I_d) = \Sigma'$. The inverse of the exponential map is then $\log_\Sigma \Sigma' = T_{\Sigma \to \Sigma'} - I_d$.

The description of the Bures-Wasserstein tangent space is in accordance with the general Riemannian structure of Wasserstein space (see [AGS08]) and agrees with the convention in [Che+20]. We now elaborate on other possible conventions, in order to dispel possible confusion.

The space $\mathbb{S}_{++}^d$ is often studied as a manifold in other contexts, and the tangent space at any point is usually identified with $\mathbb{S}^d$. It is crucial to realize, however, that a tangent space is not simply a vector space (or inner product space); a tangent space also has the interpretation of describing velocities of curves. In other words, for each tangent vector $S$, we also need to prescribe which curves have velocity $S$. In the usual way of describing the manifold structure of $\mathbb{S}_{++}^d$, this prescription is given as follows. Given a curve $(\Sigma_t)_{t \in \mathbb{R}} \subseteq \mathbb{S}_{++}^d$, if $\dot{\Sigma}_0$ denotes the ordinary time derivative of this curve at time 0, then we declare $\dot{\Sigma}_0$ to be the tangent vector of the curve at time 0. Although this prescription is natural, observe that it conflicts with our description of the tangent space structure of the Bures-Wasserstein manifold; in particular, for the curve in (10), we have described the tangent vector to this curve (at time 0) to be $T_{\Sigma \to \Sigma'} - I_d$, but the ordinary time derivative of this curve is $(T_{\Sigma \to \Sigma'} - I_d)\Sigma + \Sigma(T_{\Sigma \to \Sigma'} - I_d)$.

To summarize the discussion in the preceding paragraph: although the usual description of the tangent space of $\mathbb{S}_{++}^d$ at $\Sigma$ and our description of the tangent space are formally the same, in that they are both identified with $\mathbb{S}^d$, they differ in that tangent vectors from the two descriptions give rise to different curves. Note that if we were to adopt the usual description of the tangent space of $\mathbb{S}_{++}^d$, then we would have to define the tangent space norm $\|\cdot\|_\Sigma$ differently from above. In this paper, we adopt the convention described earlier in this section in order to preserve the connection with the general setting of optimal transport.

## A.2 Geodesic convexity and generalized geodesic convexity

Once we have geodesics, we can then define convex functions. A function $f : \mathcal{P}_{2,\mathrm{ac}}(\mathbb{R}^d) \to \mathbb{R}$ is said to be *geodesically convex* if for all constant-speed geodesics $(\mu_t)_{t \in [0,1]}$ (i.e., curves described by (8)), it holds that

$$f(\mu_t) \le (1-t)\,f(\mu_0) + t\,f(\mu_1), \qquad t \in [0,1]. \tag{11}$$

It turns out, however, that many natural examples of geodesically convex functions on Wasserstein space are convex in a stronger sense, in that they satisfy the inequality (11) for a larger class of curves than geodesics. A *generalized geodesic* from $\mu_0$ to $\mu_1$, with basepoint $\nu \in \mathcal{P}_{2,\mathrm{ac}}(\mathbb{R}^d)$, is defined to be the curve $(\mu_t^\nu)_{t \in [0,1]}$ defined by

$$\mu_t^\nu := [(1-t)T_{\nu \to \mu_0} + tT_{\nu \to \mu_1}]_\# \nu, \qquad t \in [0,1].$$

A function $f : \mathcal{P}_{2,\mathrm{ac}}(\mathbb{R}^d) \to \mathbb{R}$ is said to be *convex along generalized geodesics* if for every generalized geodesic $(\mu_t^\nu)_{t \in [0,1]}$, it holds that

$$f(\mu_t^\nu) \le (1-t)\,f(\mu_0) + t\,f(\mu_1), \qquad t \in [0,1].$$

Note that the geodesic $(\mu_t)_{t \in [0,1]}$ joining $\mu_0$ to $\mu_1$ coincides with the generalized geodesic $(\mu_t^{\mu_0})_{t \in [0,1]}$, so that convexity along generalized geodesics is indeed stronger than geodesic convexity.

Generalized geodesics were studied in [AGS08] in order to rigorously study gradient flows on Wasserstein space. The added flexibility of generalized geodesics is sometimes important for applications [AC20]; in our work, as well as in [Che+20], generalized geodesics are needed to study the iterates of Riemannian GD.

The interpretation of generalized geodesics is that we linearize $\mathcal{P}_{2,\mathrm{ac}}(\mathbb{R}^d)$ on the tangent space $T_\nu \mathcal{P}_{2,\mathrm{ac}}(\mathbb{R}^d)$. This means that we replace $\mu_0$ with its image $\log_\nu \mu_0 = T_{\nu \to \mu_0} - \mathrm{id}$ in the tangent

---

[4]Technically the exponential map is only defined if $S + I_d \succeq 0$; this is because if $S + I_d$ is not positive semidefinite, then $S + I_d$ is not an optimal transport map due to Brenier's theorem.

space, and similarly for $\mu_1$. Since the tangent space is a subset of a Hilbert space, geodesics in the tangent space are described by straight lines, i.e.,

$$t \mapsto (1-t)T_{\nu \to \mu_0} + tT_{\nu \to \mu_1} - \mathrm{id} \,.$$

If we translate back to $\mathcal{P}_{2,\mathrm{ac}}(\mathbb{R}^d)$, we end up with the curve

$$t \mapsto \exp_\nu\big((1-t)T_{\nu \to \mu_0} + tT_{\nu \to \mu_1} - \mathrm{id}\big) = [(1-t)T_{\nu \to \mu_0} + tT_{\nu \to \mu_1}]_\# \nu = \mu_t^\nu \,.$$

Thus, the property of being convex along generalized geodesics can be reformulated as requiring that

$$f \circ \exp_\nu : T_\nu \mathcal{P}_{2,\mathrm{ac}}(\mathbb{R}^d) \to \mathbb{R} \qquad \text{is convex for every } \nu \in \mathcal{P}_{2,\mathrm{ac}}(\mathbb{R}^d)\,. \tag{12}$$

In Euclidean space, convexity of a function $f : \mathbb{R}^d \to \mathbb{R}$ is equivalent, via Jensen's inequality, to the following statement: for every probability measure $P$ on $\mathbb{R}^d$, it holds that $f(\int x \, \mathrm{d}P(x)) \le \int f(x) \, \mathrm{d}P(x)$. Since the Wasserstein barycenter is the Wasserstein analogue of the mean, we can write a similar definition on Wasserstein space. Given a probability measure $P$ on $\mathcal{P}_{2,\mathrm{ac}}(\mathbb{R}^d)$, let $b_P$ denote its Wasserstein barycenter. We say that $f : \mathcal{P}_{2,\mathrm{ac}}(\mathbb{R}^d) \to \mathbb{R}$ is *convex along barycenters* if

$$f(b_P) \le \int f(\mu) \, \mathrm{d}P(\mu)\,, \qquad \text{for all } P \in \mathcal{P}_2\big(\mathcal{P}_{2,\mathrm{ac}}(\mathbb{R}^d)\big)\,.$$

Similarly, via (12), we can define $f : \mathcal{P}_{2,\mathrm{ac}}(\mathbb{R}^d) \to \mathbb{R}$ to be convex along generalized barycenters if

$$f \circ \exp_\nu\Big(\int v \, \mathrm{d}P(v)\Big) \le \int f \circ \exp_\nu(v) \, \mathrm{d}P(v)$$
$$\text{for all } \nu \in \mathcal{P}_{2,\mathrm{ac}}(\mathbb{R}^d) \text{ and } P \in \mathcal{P}_2\big(T_\nu \mathcal{P}_{2,\mathrm{ac}}(\mathbb{R}^d)\big)\,. \tag{13}$$

However, since the tangent space is embedded in a Hilbert space, there is no difference between (12) and (13).

To summarize the relationship between these four concepts:

convex along generalized barycenters $\iff$ convex along generalized geodesics
$$\implies \text{ convex along barycenters } \implies \text{ geodesically convex}\,.$$

For a justification of these facts and further discussion, see [AC11].

## A.3 Geodesic optimization

Given a functional $F : \mathcal{P}_{2,\mathrm{ac}}(\mathbb{R}^d) \to \mathbb{R}$, we can define its Wasserstein gradient formally as follows. For any constant-speed geodesic $(\mu_t)_{t \in [0,1]}$, the gradient of $F$ at $\mu_0$ is the element $\nabla F(\mu_0) \in T_{\mu_0} \mathcal{P}_{2,\mathrm{ac}}(\mathbb{R}^d)$ satisfying

$$\partial_t|_{t=0} F(\mu_t) = \langle \nabla F(\mu_0), T_{\mu_0 \to \mu_1} - \mathrm{id} \rangle_{\mu_0}\,.$$

The Riemannian GD update for $F$ with step size $\eta$ starting at $\mu$ is

$$\mu^+ := \exp_\mu\big(-\eta \nabla F(\mu)\big) = [\mathrm{id} - \eta \nabla F(\mu)]_\# \mu\,.$$

Note that the step size $\eta$ should be chosen small enough that $-\eta \nabla F(\mu)$ lies in the domain of the exponential map. From the general description of the tangent space of Wasserstein space, $\nabla F(\mu)$ is the gradient of a mapping $\psi : \mathbb{R}^d \to \mathbb{R}$; then, $-\eta \nabla F(\mu)$ belongs to the domain of the exponential map if $\|\cdot\|^2/2 - \eta\psi$ is convex.

We say that $F$ is $\alpha$-strongly convex if

$$F(\mu_1) \ge F(\mu_0) + \langle \nabla F(\mu_0), \log_{\mu_0} \mu_1 \rangle_{\mu_0} + \frac{\alpha}{2} W_2^2(\mu_0, \mu_1)\,, \qquad \text{for all } \mu_0, \mu_1 \in \mathcal{P}_{2,\mathrm{ac}}(\mathbb{R}^d)\,,$$

and $\beta$-smooth if

$$F(\mu_1) \le F(\mu_0) + \langle \nabla F(\mu_0), \log_{\mu_0} \mu_1 \rangle_{\mu_0} + \frac{\beta}{2} W_2^2(\mu_0, \mu_1)\,, \qquad \text{for all } \mu_0, \mu_1 \in \mathcal{P}_{2,\mathrm{ac}}(\mathbb{R}^d)\,.$$

These two properties are formally equivalent to the following statements: for any constant-speed geodesic $(\mu_t)_{t \in [0,1]}$, one has

$$\partial_t^2|_{t=0} F(\mu_t) \ge \alpha \, W_2^2(\mu_0, \mu_1) \qquad \text{or} \qquad \partial_t^2|_{t=0} F(\mu_t) \le \beta \, W_2^2(\mu_0, \mu_1)\,,$$

respectively.

## A.4 Curvature and the barycenter functional

One of the interesting features of the barycenter problem is that, because it is defined in terms of the squared distance function, it captures key geometric features of the underlying space; in fact, this is arguably the reason for the success of the barycenter for geometric applications. To further discuss this connection, it is insightful to abstract the situation to computing barycenters on a metric space.

Given a metric space $(X, d)$ and a probability measure $P$ on $X$, a barycenter of $P$ is a solution of

$$\underset{b \in X}{\text{minimize}} \qquad F_P(b) := \frac{1}{2} \int d^2(b, \cdot) \, dP \, .$$

The basic structure required on $X$ in order to study first-order optimization methods is the presence of geodesics. This is formalized by the notion of a *geodesic space*, which is studied in metric geometry; see [BBI01]. Then, we may define a function $F : X \to \mathbb{R}$ to be $\alpha$-*strongly convex* if for all geodesics $(x_t)_{t \in [0,1]}$ in $X$, it holds that

$$F(x_t) \leq (1 - t) \, F(x_0) + t \, F(x_1) - \frac{\alpha \, t \, (1 - t)}{2} \, d^2(x_0, x_1) \, , \qquad \text{for all } t \in [0, 1] \, .$$

It is known that the convexity properties of the barycenter functional $F_P$ are related to the *curvature* of the space. Here, curvature is interpreted as the *Alexandrov curvature*, which is the generalization of sectional curvature to geodesic spaces, see [BBI01]. Then, the result is that $F_P$ is 1-strongly convex for every probability measure $P$ on $X$ if and only if $X$ has *non-positive curvature*; see [Stu03] for precise statements. In fact, the 1-strong convexity of barycenter functionals is essentially the definition of non-positive curvature in this context.

Consequently, much stronger results are known for barycenters in non-positively curved spaces, ranging from basic properties such as existence and uniqueness, to statistical estimation and optimization; for details see the nice article [Stu03].

In contrast, it is well-known that Wasserstein space $\mathcal{P}_{2,\text{ac}}(\mathbb{R}^d)$ (and hence, the Bures-Wasserstein space) is *non-negatively curved* [AGS08, Theorem 7.3.2]. This means, for instance, that convexity and properties related to convexity (such as the PL inequality employed in Appendix C.1) are not automatic for the barycenter functional in Wasserstein space. On the other hand, as emphasized in [Che+20], this non-negative curvature is related to the *smoothness* of the barycenter functional.

## A.5 Additional facts about the Wasserstein metric

Here we collect various facts about the Wasserstein metric for easy reference in the sequel.

1. **Euclidean gradient vs. Bures-Wasserstein gradient.**
   Let $F : \mathbb{S}_{++}^d \to \mathbb{R}$ be a function. Throughout this paper, we denote by $\mathrm{D} \, F$ the usual Euclidean gradient of $F$, and we reserve $\nabla F$ for the gradient with respect to the Bures-Wasserstein geometry. In fact, under our tangent space convention, these two quantities are related as follows: let $(\Sigma_t)_{t \in \mathbb{R}}$ denote a curve in $\mathbb{S}_{++}^d$. We temporarily denote the Euclidean tangent vector (i.e., ordinary time derivative) to this curve via $\dot{\Sigma}^{\mathrm{E}}$, and the Bures-Wasserstein tangent vector via $\dot{\Sigma}^{\mathrm{BW}}$, which are related via $\dot{\Sigma}^{\mathrm{E}} = \dot{\Sigma}^{\mathrm{BW}} \Sigma + \Sigma \dot{\Sigma}^{\mathrm{BW}}$ (see the discussion in Appendix A.1). We can compute the time derivative of $F$ in two ways:

$$\langle \nabla F(\Sigma_0), \dot{\Sigma}_0^{\mathrm{BW}} \rangle_{\Sigma_0} = \partial_t|_{t=0} F(\Sigma_t) = \langle \mathrm{D} \, F(\Sigma_0), \dot{\Sigma}_0^{\mathrm{E}} \rangle = \langle \mathrm{D} \, F(\Sigma_0), \dot{\Sigma}_0^{\mathrm{BW}} \Sigma_0 + \Sigma_0 \dot{\Sigma}_0^{\mathrm{BW}} \rangle$$
$$= 2 \, \langle \mathrm{D} \, F(\Sigma_0), \dot{\Sigma}_0^{\mathrm{BW}} \rangle_{\Sigma_0} \, .$$

   From this we can conclude that

$$\nabla F(\Sigma_0) = 2 \, \mathrm{D} \, F(\Sigma_0) \, .$$

2. **Gradient of the squared Wasserstein distance.**
   For any $\nu \in \mathcal{P}_{2,\text{ac}}(\mathbb{R}^d)$, the gradient of the functional $W_2^2(\cdot, \nu)$ at $\mu$ is given by

$$\nabla W_2^2(\cdot, \nu)(\mu) = -2 \, (T_{\mu \to \nu} - \mathrm{id}) = -2 \log_\mu \nu \, .$$

   This is derived in, e.g. [ZP19]. In the Bures-Wasserstein setting, it can be proven via matrix calculus; see the proof of Theorem 3.

3. **Inverse of the transport map**.

   If $\Sigma, \Sigma' \in \mathbb{S}_{++}^d$, then the transport map $T_{\Sigma \to \Sigma'}$ is the inverse matrix for the transport map $T_{\Sigma' \to \Sigma}$. This can be verified from the formula (6) using the symmetry of the geometric mean. More generally, it is a special case of the convex conjugacy relation between optimal Kantorovich potentials.

4. **Diagonal case**.

   If $\Sigma_0, \Sigma_1 \in \mathbb{S}_{++}^d$ are *diagonal matrices*, then $W_2^2(\Sigma_0, \Sigma_1) = \|\Sigma_0^{1/2} - \Sigma_1^{1/2}\|_{\mathrm{F}}^2$ is the squared Frobenius norm between the square roots. This can be verified, e.g. from the explicit formula (5) using the fact that $\Sigma_0$ and $\Sigma_1$ commute. Note that in one dimension, all matrices are diagonal. More generally, these observations extend to when $\Sigma_0$ and $\Sigma_1$ commute.

   Similarly, it can be seen from (10) that the geodesic is given by

   $$\Sigma_t^{1/2} = (1 - t)\,\Sigma_0^{1/2} + t\,\Sigma_1^{1/2}\,, \qquad t \in [0, 1]\,,$$

   which says that the Bures-Wasserstein geodesic between diagonal (or commuting matrices) is simply the Euclidean geodesic after applying the square root map.

5. **The case of non-zero means**.

   For any $\mu, \nu \in \mathcal{P}_2(\mathbb{R}^d)$, suppose that the means of these distributions are $m_\mu$ and $m_\nu$, respectively. Let $\bar\mu, \bar\nu$ denote the centered versions of these distributions. Then, it holds that

   $$W_2^2(\mu, \nu) = \|m_\mu - m_\nu\|^2 + W_2^2(\bar\mu, \bar\nu)\,.$$

   This can be proven directly from the definition (4).

6. **A lower bound on the Wasserstein distance**.

   Let $\mu, \nu \in \mathcal{P}_2(\mathbb{R}^d)$. If $\tilde\mu$ and $\tilde\nu$ are *Gaussian* measures with the same moments up to order two as $\mu$ and $\nu$, respectively, then $W_2(\mu, \nu) \geq W_2(\tilde\mu, \tilde\nu)$ [CMT96].

## B  Proofs for the geodesic convexity results

### B.1  Proof of Theorem 1

See Appendix A.1 and A.2 for background on the relevant geometric concepts.

*Proof of Theorem 1.* We begin by proving that $-\sqrt{\lambda_{\min}}$ is convex (equivalently, $\sqrt{\lambda_{\min}}$ is concave) along generalized geodesics. Let $Q$ be any finitely supported probability measure on $\mathbb{S}_{++}^d$, and let $\Sigma_0 \in \mathbb{S}_{++}^d$ denote the basepoint. It is equivalent to show that if $\Sigma^\star$ is the generalized barycenter of $Q$ at $\Sigma_0$, then $\sqrt{\lambda_{\min}(\Sigma^\star)} \geq \int \sqrt{\lambda_{\min}(\Sigma)}\,\mathrm{d}Q(\Sigma)$.

The generalized barycenter by definition is the matrix $\Sigma^\star = \exp_{\Sigma_0}(\int T_{\Sigma_0 \to \Sigma}\,\mathrm{d}Q(\Sigma) - I_d)$. If we write $\bar T := \int T_{\Sigma_0 \to \Sigma}\,\mathrm{d}Q(\Sigma)$ for the average transport map, the statement we want to show is $\bar T \Sigma_0 \bar T \succeq \alpha I_d$ where $\alpha := (\int \sqrt{\lambda_{\min}(\Sigma)}\,\mathrm{d}Q(\Sigma))^2$. We observe that

$$\bar T \Sigma_0 \bar T \succeq \alpha I_d \iff \bar T^{-1}\Sigma_0^{-1}\bar T^{-1} \preceq \alpha^{-1} I_d \iff \bar T^{-1} \preceq \mathrm{GM}(\Sigma_0, \alpha^{-1}I_d)\,,$$

where the first equivalence follows from the order preservation of inversion [BL06, Exercise 3.3.2] and the second from [LL01, Corollary 3.5]. In turn, this is equivalent to $\bar T \succeq \alpha^{1/2}\Sigma_0^{-1/2}$.

Since $\bar T = \int \Sigma_0^{-1/2}(\Sigma_0^{1/2}\Sigma\Sigma_0^{1/2})^{1/2}\Sigma_0^{-1/2}\,\mathrm{d}Q(\Sigma)$, we want to prove $\int (\Sigma_0^{1/2}\Sigma\Sigma_0^{1/2})^{1/2}\,\mathrm{d}Q(\Sigma) \succeq \alpha^{1/2}\Sigma_0^{1/2}$. To prove this, observe that $\Sigma \succeq \lambda_{\min}(\Sigma)\,I_d$, so $\Sigma_0^{1/2}\Sigma\Sigma_0^{1/2} \succeq \lambda_{\min}(\Sigma)\,\Sigma_0$. Since taking square roots preserves the PSD ordering (c.f. [BL06, Exercise 1.2.5]), upon taking square roots and integrating we deduce

$$\int (\Sigma_0^{1/2}\Sigma\Sigma_0^{1/2})^{1/2}\,\mathrm{d}Q(\Sigma) \succeq \left(\int \sqrt{\lambda_{\min}(\Sigma)}\,\mathrm{d}Q(\Sigma)\right)\Sigma_0^{1/2} = \alpha\Sigma_0^{1/2}\,.$$

Hence $-\sqrt{\lambda_{\min}}$ is convex along generalized geodesics.

The proof of convexity of $\sqrt{\lambda_{\max}}$ is similar. By [LL01, Corollary 3.5],

$$\bar T \Sigma_0 \bar T \preceq \beta I_d \iff \bar T \preceq \mathrm{GM}(\Sigma_0^{-1}, \beta I_d)\,.$$

Since $\bar{T} = \int \Sigma_0^{-1/2} (\Sigma_0^{1/2}\Sigma\Sigma_0^{1/2})^{1/2} \Sigma_0^{-1/2} \, dQ(\Sigma)$, it thus suffices to show $\int (\Sigma_0^{1/2}\Sigma\Sigma_0^{1/2})^{1/2} \, dQ(\Sigma) \preceq \beta^{1/2}\Sigma_0^{1/2}$ where $\beta := \left(\int \sqrt{\lambda_{\max}(\Sigma)} \, dQ(\Sigma)\right)^2$ as desired. Noting that $\Sigma_0^{1/2}\Sigma\Sigma_0^{1/2} \preceq \lambda_{\max}(\Sigma)\,\Sigma_0$, taking square roots and integrating yields

$$\int (\Sigma_0^{1/2}\Sigma\Sigma_0^{1/2})^{1/2} \, dQ(\Sigma) \preceq \left(\int \sqrt{\lambda_{\max}(\Sigma)} \, dQ(\Sigma)\right) \Sigma_0^{1/2} = \beta\Sigma_0^{1/2}.$$

Hence the result. □

*Remark* 2. This result implies for instance that the set of PSD matrices with eigenvalues lying in a certain range is convex along generalized geodesics.

*Remark* 3. There is a short proof of the weaker statement that the functionals $-\sqrt{\lambda_{\min}}$ and $\sqrt{\lambda_{\max}}$ are geodesically convex. The following argument is implicit in the proofs of [AC11, Theorem 6.1] and [BJL19, Theorem 8]. Let $Q$ be a finitely supported probability measure on $\mathbb{S}_{++}^d$. The barycenter $\Sigma^\star$ of $Q$ satisfies the fixed-point equation

$$\Sigma^\star = \int (\Sigma^{\star 1/2}\Sigma\Sigma^{\star 1/2})^{1/2} \, dQ(\Sigma),$$

see [AC11, Theorem 6.1]. This implies

$$\lambda_{\min}(\Sigma^\star) \geq \int \sqrt{\lambda_{\min}(\Sigma^{\star 1/2}\Sigma\Sigma^{\star 1/2})} \, dQ(\Sigma) \geq \sqrt{\lambda_{\min}(\Sigma^\star)} \int \sqrt{\lambda_{\min}(\Sigma)} \, dQ(\Sigma).$$

A similar argument applies for $\sqrt{\lambda_{\max}}$.

### B.2 Sharpness of Theorem 1

We investigate the sharpness of this result in the following sense: for what exponents $p \geq 0$ is it true that the functionals $-\lambda_{\min}^p$, $\lambda_{\max}^p$ are geodesically convex? For instance, the functional $\lambda_{\max}$ was shown to be geodesically convex in [Che+20, Lemma 13].

In the following theorem, we show that the exponent $p = 1/2$ in Theorem 1 is optimal, in the sense that all possible geodesic convexity statements involving powers of $\lambda_{\min}$ and $\lambda_{\max}$ (except the trivial case $p = 0$) can be deduced from our result for $p = 1/2$.

**Theorem 6.** *The following diagrams depict the exponents $p \in \mathbb{R}$ for which $\lambda_{\min}^p$ and $\lambda_{\max}^p$ are concave or convex.*

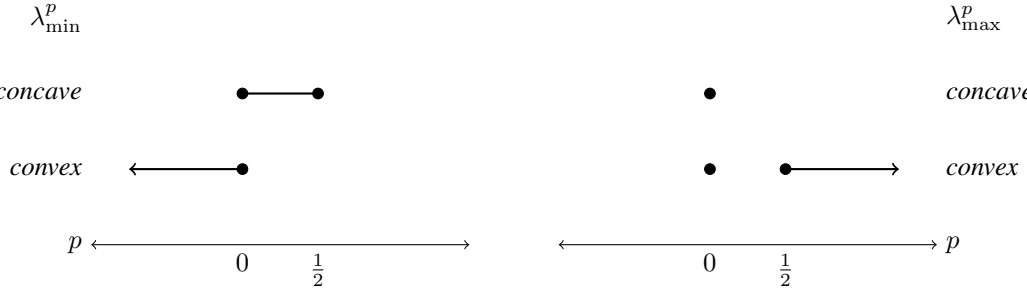

*The diagram is to be interpreted as follows. If part of the diagram is filled in with a solid black line, then the corresponding functional is concave/convex along generalized geodesics (and hence it is geodesically concave/convex). If part of the diagram is* not *filled in, then there exist counterexamples showing that the functional is* not *geodesically concave/convex.*

*Proof.* First, we establish the positive results, which follow from composition rules:

- For $0 \leq p \leq 1/2$, $\lambda_{\min}^p$ is the composition of the increasing concave function $(\cdot)^{2p}$ with the concave function $\sqrt{\lambda_{\min}}$, so it is concave.

- For $p \leq 0$, $\lambda_{\min}^p$ is the composition of the decreasing convex function $(\cdot)^{2p}$ with the concave function $\sqrt{\lambda_{\min}}$, so it is convex.

- For $p \geq 1/2$, $\lambda_{\max}^p$ is the composition of the increasing convex function $(\cdot)^{2p}$ with the convex function $\sqrt{\lambda_{\max}}$, so it is convex.

Next, we turn towards the negative results. First, recall from Fact 4 in Appendix A.5 that if $\Sigma_0$ and $\Sigma_1$ are one-dimensional, i.e., they are positive numbers, then the Bures-Wasserstein geodesic is

$$\Sigma_t = \left((1-t)\Sigma_0^{1/2} + t\Sigma_1^{1/2}\right)^2, \qquad t \in [0,1].$$

Also, in this case, $\lambda_{\min}$ and $\lambda_{\max}$ coincide and equal the identity; we thus abuse notation slightly in this paragraph by writing $\lambda$ for both to handle the two cases simultaneously. Once we reparametrize by the square roots, it is seen that asking for concavity/convexity of $\lambda^p$ is equivalent to asking for usual convexity of $(\cdot)^{2p}$ on $\mathbb{R}_+$. This example rules out: (1) the concavity of $\lambda^p$ for $p < 0$; (2) the convexity of $\lambda^p$ for $0 < p < 1/2$; and (3) the concavity of $\lambda^p$ for $p > 1/2$.

To rule out convexity of $\lambda_{\min}^p$ for $p > 0$, consider $\Sigma = \mathrm{diag}(\varepsilon, 1/\varepsilon)$ for small $\varepsilon > 0$. The transport map from $\Sigma^{-1}$ to $\Sigma$ is $\Sigma$, so from (10) the midpoint of this geodesic is $M := (\Sigma + \Sigma^{-1} + 2I_2)/4 = (\varepsilon + \varepsilon^{-1} + 2)I_2/4$. In particular, this implies that $\lambda_{\min}(M) \geq 1/(4\varepsilon) \gg \varepsilon = \max\{\lambda_{\min}(\Sigma), \lambda_{\min}(\Sigma^{-1})\}$. Thus $\lambda_{\min}^p$ is not convex for any $p > 0$.

To rule out concavity of $\lambda_{\max}^p$ for $p > 0$, note that for $\varepsilon$ sufficiently small, in the previous example $\lambda_{\max}(M) \approx 1/(4\varepsilon) \ll 1/\varepsilon = \max\{\lambda_{\max}(\Sigma), \lambda_{\max}(\Sigma^{-1})\}$. Also, for any $p < 0$, the convexity of $\lambda_{\max}^p$ would imply the concavity of $\lambda_{\max}^{-p}$ due to the composition rules, hence $\lambda_{\max}^p$ is not convex.

This covers all cases. $\qquad\square$

### B.3 Eigenvalue clipping is a Bures-Wasserstein contraction

Convex sets play an important role in Euclidean optimization because projection onto a convex set is a contraction (c.f. [Bub15, Lemma 3.1]), and hence projected GD can be used to solve constrained optimization. Unfortunately, as the Bures-Wasserstein space is positively curved, we cannot automatically conclude that projection onto a geodesically convex set is a projection. Nevertheless, we can verify by hand the following result. In what follows, define for $0 < \beta < \infty$ the operator $\mathrm{clip}^\beta : \mathbb{S}_{++}^d \to \mathbb{S}_{++}^d$ in the following way: if $\Sigma = \sum_{i=1}^d \lambda_i u_i u_i^\mathsf{T}$ is an eigenvalue decomposition of $\Sigma$, then

$$\mathrm{clip}^\beta \Sigma := \sum_{i=1}^d (\lambda_i \wedge \beta)\, u_i u_i^\mathsf{T}.$$

**Proposition 3.** *The operator* $\mathrm{clip}^\beta$ *is a contraction w.r.t. the Bures-Wasserstein metric, i.e.,* $W_2(\mathrm{clip}^\beta \Sigma, \mathrm{clip}^\beta \Sigma') \leq W_2(\Sigma, \Sigma')$.

To prove this proposition, we first extend the clipping operation to an operator $\mathbb{R}^{d \times d} \to \mathbb{R}^{d \times d}$ via the singular values; namely, given a singular value decomposition $A = \sum_{i=1}^d s_i u_i v_i^\mathsf{T}$, we let $\mathrm{clip}^\beta A := \sum_{i=1}^d (s_i \wedge \beta)\, u_i v_i^\mathsf{T}$.

*Proof of Proposition 3.* Fix $X, Y \in \mathbb{S}_{++}^d$. It is known (see e.g. [BJL19]) that

$$W_2(X, Y) = \min_{\substack{A, B \in \mathbb{R}^{d \times d} \\ AA^\mathsf{T} = X \\ BB^\mathsf{T} = Y}} \|A - B\|_\mathrm{F}.$$

Let $(\bar{A}, \bar{B})$ be a minimizing pair in the above expression. We aim to show

$$W_2(\mathrm{clip}^\beta X, \mathrm{clip}^\beta Y) \leq \|\mathrm{clip}^{\sqrt{\beta}} \bar{A} - \mathrm{clip}^{\sqrt{\beta}} \bar{B}\|_\mathrm{F} \overset{?}{\leq} \|\bar{A} - \bar{B}\|_\mathrm{F} = W_2(X, Y).$$

We only have to show the second inequality, and we do so by showing that the operator $\mathrm{clip}^M : \mathbb{R}^{d \times d} \to \mathbb{R}^{d \times d}$ satisfies

$$\mathrm{clip}^M A = \underset{\tilde{A} \in \mathbb{R}^{d \times d},\ \|\tilde{A}\| \leq M}{\arg\min} \|A - \tilde{A}\|_\mathrm{F}, \qquad A \in \mathbb{R}^{d \times d}. \tag{14}$$

This will prove that $\text{clip}^M$ is the Euclidean *projection* onto the closed convex set $\{\|\cdot\| \leq M\}$, and such a projection is automatically 1-Lipschitz.

Indeed, showing (14) is standard. Write $A = U\Sigma V^\mathsf{T}$ for its singular value decomposition.

$$\underset{\tilde{A} \in \mathbb{R}^{d \times d}, \|\tilde{A}\| \leq M}{\arg\min} \|\tilde{A} - A\|_\mathrm{F}^2 = \underset{\tilde{A} \in \mathbb{R}^{d \times d}, \|\tilde{A}\| \leq M}{\arg\min} \|\tilde{A} - U\Sigma V^\mathsf{T}\|_\mathrm{F}^2 = \underset{\tilde{A} \in \mathbb{R}^{d \times d}, \|\tilde{A}\| \leq M}{\arg\min} \|U^\mathsf{T}\tilde{A}V - \Sigma\|_\mathrm{F}^2$$

$$= \underset{\tilde{A} \in \mathbb{R}^{d \times d}, \|\tilde{A}\| \leq M}{\arg\min} \left\{ \sum_{i=1}^d \left\{\Sigma[i,i] - (U^\mathsf{T}\tilde{A}V)[i,i]\right\}^2 + \sum_{\substack{i,j\in[d]\\i\neq j}} (U^\mathsf{T}\tilde{A}V)[i,j]^2 \right\}.$$

On the other hand,

$$\underset{\tilde{A} \in \mathbb{R}^{d \times d}, \|\tilde{A}\| \leq M}{\min} \left\{ \sum_{i=1}^d \left\{\Sigma[i,i] - (U^\mathsf{T}\tilde{A}V)[i,i]\right\}^2 + \sum_{i,j\in[d],\, i\neq j} (U^\mathsf{T}\tilde{A}V)[i,j]^2 \right\}$$

$$\geq \sum_{i=1}^d \left\{(\Sigma[i,i] - M)_+\right\}^2,$$

with equality attained at the unique minimizer $\tilde{A}$ satisfying $U^\mathsf{T}\tilde{A}V = \text{clip}^M\Sigma$, i.e., $\tilde{A} = \text{clip}^M A$.

$\square$

## C  Proofs for barycenters

### C.1  Riemannian gradient descent

In this section, we detail the obstacles faced by previous analyses and then show how our geodesic convexity result, Theorem 1, enables us to overcome the prior exponential dependence on dimension and obtain the dimension-free rates in Theorems 2 and 3.

We begin by recalling the proof strategy of [Che+20]. Let $F$ denote the barycenter functional,

$$F(\Sigma) := \frac{1}{2} \int W_2^2(\Sigma, \cdot)\, \mathrm{d}P. \tag{15}$$

Standard optimization guarantees are often proven under the assumption that the objective function $F$ is smooth and convex. Since we are considering Riemannian descent, this should be interpreted as convex and smooth along geodesics, as in [ZS16]. Unfortunately, the functional $F$ is not geodesically convex (see [Che+20, Appendix B.2]), and so we must look for weaker conditions which still imply convergence of GD/SGD. A gradient domination condition known as the *Polyak-Łojasiewicz inequality* (henceforth *PL inequality*) was introduced in the non-convex optimization literature as an appropriate substitute for strong convexity [Pol64; Loj63], see also e.g., [KNS16; Bol+17]). Establishing a PL inequality in the present setting plays a key role in the analysis.

The following properties of the barycenter functional were proven in [Che+20].

**Theorem 7.** *Let $0 < \lambda_{\min} \leq \lambda_{\max} < \infty$ and write $\kappa := \lambda_{\max}/\lambda_{\min}$.*

1. *([Che+20, Theorem 7]) The barycenter functional $F$ is 1-geodesically smooth.*

2. *([Che+20, Theorem 17]) Assume that the covariance matrices in the support of $P$ have eigenvalues in the range $[\lambda_{\min}, \lambda_{\max}]$. Then, $F$ satisfies a* variance inequality,

$$F(\Sigma) - F(\Sigma^\star) \geq \frac{1}{2\kappa} W_2^2(\Sigma, \Sigma^\star), \qquad \text{for all } \Sigma \in \mathbb{S}_{++}^d.$$

3. *([Che+20, Theorem 19]) Assume that the covariance matrices in the support of $P$, as well as $\Sigma$ itself, have eigenvalues in the range $[\lambda_{\min}, \lambda_{\max}]$. Then, $F$ satisfies a PL inequality at the matrix $\Sigma$:*

$$F(\Sigma) - F(\Sigma^\star) \leq 2\kappa^2 \|\nabla F(\Sigma)\|_\Sigma^2.$$

Geodesic smoothness together with a PL inequality at every iterate are enough to obtain convergence guarantees for GD/SGD in objective value (i.e., the quantity $F(\Sigma) - F(\Sigma^\star)$), c.f. [Che+20, Theorems 4-5]. The variance inequality is then used to deduce convergence of the iterate to $\Sigma^\star$.

The main difficulty when applying these results is the assumption required for the third point: it requires *a priori* control over the eigenvalues of the iterates of GD/SGD.

This difficulty is addressed in [Che+20] via the following strategy: identify a geodesically convex subset $\mathcal{S}$ of the Bures-Wasserstein manifold for which we can prove uniform bounds on the eigenvalues of matrices in $\mathcal{S}$. Since the iterates of SGD travel along geodesics, if $P$ is supported in $\mathcal{S}$ and the algorithm is initialized in $\mathcal{S}$, it follows that all iterates of SGD will remain in $\mathcal{S}$. The situation is similar for GD, except that "geodesics" must be replaced by "generalized geodesics".

We can now describe the source of the exponential dependence on dimension in the result of [Che+20]: if the covariance matrices in the support of $P$ have eigenvalues in $[\lambda_{\min}, \lambda_{\max}]$, then the subset $\mathcal{S}$ used in the analysis of Chewi et al. is substantially larger than the support of $P$, and in particular the eigenvalues of matrices in $\mathcal{S}$ can only be proven to lie in the range $[\lambda_{\min}/\kappa^{d-1}, \lambda_{\max}]$. The main improvement in the present analysis is to use our geodesic convexity result (Theorem 1) to prove the following result.

**Lemma 1.** *Suppose that the covariance matrices in the support of $P$ have eigenvalues in the range* $[\lambda_{\min}, \lambda_{\max}]$, *and that we initialize GD (respectively SGD) at a point in* $\operatorname{supp} P$. *Then, the iterates of GD (respectively SGD) also have eigenvalues in the range* $[\lambda_{\min}, \lambda_{\max}]$.

*Proof.* From Theorem 1, the set of matrices with eigenvalues in $[\lambda_{\min}, \lambda_{\max}]$ is closed under generalized geodesics. Since the update of GD (respectively SGD) moves along generalized geodesics (respectively geodesics), the result follows. $\square$

This combined with the arguments below is enough to alleviate the exponential dimension dependence. However, before continuing to the main argument, we prove sharper bounds for the last two statements of Theorem 7. This allows us to also improve our convergence rates' dependence on the conditioning.

This improved version of Theorem 7 rests on the following observation. [Che+20, Lemma 16] shows that if $\Sigma$, $\Sigma'$ have eigenvalues which lie in the range $[\lambda_{\min}, \lambda_{\max}]$, then the eigenvalues of the transport map $T_{\Sigma \to \Sigma'}$ lie in $[\kappa^{-1}, \kappa]$. However, these bounds are loose, as following lemma shows.

**Lemma 2.** *Suppose that $\Sigma, \Sigma' \in \mathbb{S}_{++}^d$ have eigenvalues which lie in the range $[\lambda_{\min}, \lambda_{\max}]$, and let $\kappa := \lambda_{\max}/\lambda_{\min}$ denote the condition number. Then, the eigenvalues of the transport map $T_{\Sigma \to \Sigma'}$ lie in the range $[1/\sqrt{\kappa}, \sqrt{\kappa}]$.*

*Proof.* The transport map $T_{\Sigma \to \Sigma'}$ is explicitly given in (6), and it can be recognized as the matrix geometric mean of $\Sigma^{-1}$ and $\Sigma'$. Applying a norm bound for the matrix geometric mean [BG12, Theorem 3], we deduce that

$$\lambda_{\max}(T_{\Sigma \to \Sigma'}) \leq \lambda_{\max}(\Sigma'^{1/4} \Sigma^{-1/2} \Sigma'^{1/4}) \leq \sqrt{\kappa}.$$

The symmetry of $\Sigma$ and $\Sigma'$ together with Fact 3 in Appendix A.5 yields $\lambda_{\min}(T_{\Sigma \to \Sigma'}) \geq 1/\sqrt{\kappa}$. $\square$

Using this lemma, we now state and prove the refinement of Theorem 7.

**Theorem 8.** *Let $0 < \lambda_{\min} \leq \lambda_{\max} < \infty$ and write $\kappa := \lambda_{\max}/\lambda_{\min}$.*

1. *([Che+20, Theorem 7]) The barycenter functional $F$ is $1$-geodesically smooth.*

2. *Assume that the covariance matrices in the support of $P$ have eigenvalues in the range* $[\lambda_{\min}, \lambda_{\max}]$. *Then, $F$ satisfies a* variance inequality,

$$F(\Sigma) - F(\Sigma^\star) \geq \frac{1}{2\sqrt{\kappa}} W_2^2(\Sigma, \Sigma^\star), \qquad \text{for all } \Sigma \in \mathbb{S}_{++}^d.$$

3. *Assume that the covariance matrices in the support of $P$, as well as $\Sigma$ itself, have eigenvalues in the range $[\lambda_{\min}, \lambda_{\max}]$. Then, $F$ satisfies a PL inequality at the matrix $\Sigma$:*

$$F(\Sigma) - F(\Sigma^\star) \leq 2\kappa^{3/2} \|\nabla F(\Sigma)\|_\Sigma^2.$$

*Proof.* The second statement follows from the general variance inequality ([Che+20, Theorem 6]) together with Lemma 2. Similarly, the third statement follows from the proof of [Che+20, Theorem 19] using the improved variance inequality. $\qquad\square$

The proof of Theorem 2 now follows from the analysis of [Che+20, §4.2] by adapting their Theorems 4 and 5. We now sketch the modifications required to prove Theorem 3.

*Proof of Theorem 3.* We prove the stronger result that this holds for $\overline{\kappa}$, as this implies the result for $\kappa^\star$ because

$$\overline{\kappa} = \frac{\overline{\lambda_{\max}}}{\overline{\lambda_{\min}}} = \Big(\frac{\int \lambda_{\max}(\Sigma)^{1/2}\,\mathrm{d}P(\Sigma)}{\int \lambda_{\min}(\Sigma)^{1/2}\,\mathrm{d}P(\Sigma)}\Big)^2 \leq \sup_{\Sigma \in \mathrm{supp}(P)} \frac{\lambda_{\max}(\Sigma)}{\lambda_{\min}(\Sigma)} = \kappa^\star\,.$$

Above, the inequality follows from rearranging $\int \sqrt{\lambda_{\max}(\Sigma)}\,\mathrm{d}P(\Sigma) \leq \sqrt{\kappa^\star} \int \sqrt{\lambda_{\min}(\Sigma)}\,\mathrm{d}P(\Sigma)$.

We begin by checking that the variance inequality and PL inequality from Theorem 7 continue to hold under these assumptions.

**Variance inequality**. From the geodesic convexity of $-\sqrt{\lambda_{\min}}$ and $\sqrt{\lambda_{\max}}$, the barycenter $\Sigma^\star$ of $P$ has eigenvalues in $[\overline{\lambda_{\min}}, \overline{\lambda_{\max}}]$. By modifying the proof of Lemma 2 and using Fact 3 in Appendix A.5, the transport map $T_{\Sigma^\star \to \Sigma}$ has eigenvalues bounded below as

$$\lambda_{\min}(T_{\Sigma^\star \to \Sigma}) = \frac{1}{\lambda_{\max}(T_{\Sigma \to \Sigma^\star})} \geq \frac{1}{\lambda_{\max}(\Sigma^{\star\,1/4}\Sigma^{-1/2}\Sigma^{\star\,1/4})} \geq \frac{\lambda_{\min}(\Sigma)^{1/2}}{\overline{\lambda_{\max}}^{1/2}}\,.$$

From [Che+20, Theorem 6], we can deduce that the variance inequality holds for $P$ with constant

$$\int \lambda_{\min}(T_{\Sigma^\star \to \Sigma})\,\mathrm{d}P(\Sigma) \geq \frac{\overline{\lambda_{\min}}^{1/2}}{\overline{\lambda_{\max}}^{1/2}} = \frac{1}{\overline{\kappa}^{1/2}}\,.$$

**PL inequality**. Similarly, a modification of the proof of [Che+20, Theorem 19] using the improved variance inequality shows that a PL inequality holds at $\Sigma$:

$$F(\Sigma) - F(\Sigma^\star) \leq 2\overline{\kappa}^{1/2} \frac{\overline{\lambda_{\max}}}{\lambda_{\min}(\Sigma)} \|\nabla F(\Sigma)\|_\Sigma^2\,.$$

**Putting it together**. From Corollary 1, the iterates of either GD or SGD all have eigenvalues in the range $[\overline{\lambda_{\min}}, \overline{\lambda_{\max}}]$. Hence, the PL inequality in (3) of Theorem 8 holds at every iterate with $\overline{\kappa}$ replacing $\kappa$. The convergence rates now follow as before. $\qquad\square$

## C.2 Euclidean gradient descent approach

We now present our results for the Euclidean geometry. [BJL19] prove that the barycenter functional is strictly convex on the positive semidefinite cone (w.r.t. the standard Euclidean geometry). We extend their results by showing that it is in fact strongly convex and smooth (again w.r.t. the standard Euclidean geometry). Besides yielding an analysis of Euclidean projected GD and SGD, these results also aid our analysis of the regularized barycenter problem in the sequel.

Fix $0 < \alpha \leq \beta$ and denote by $\mathcal{K}_{\alpha,\beta}$ the subset of covariance matrices whose spectrum lies within $[\alpha, \beta]$. Let $F$ denote the barycenter functional, defined in (15).

**Lemma 3.** *For all $\Sigma \in \mathcal{K}_{\alpha,\beta}$ and non-zero $Y \in \mathbb{S}^d$,*

$$\frac{\alpha^3}{4\beta^4} \leq \frac{\langle Y, \mathrm{D}^2 F(\Sigma)[Y]\rangle_{\mathrm{F}}}{\|Y\|_{\mathrm{F}}^2} \leq \frac{\beta^3}{4\alpha^4}\,. \tag{16}$$

*Proof.* It suffices to consider the case where $P = \frac{1}{N}\sum_{i=1}^N \delta_{\Sigma_i}$ for some $\Sigma_i \in \mathcal{K}_{\alpha,\beta},\ i \in [N]$, as the case of general $P$ supported on $\mathcal{K}_{\alpha,\beta}$ follows by compactness. Fix $\Sigma \in \mathcal{K}_{\alpha,\beta}$. Standard calculations as in [BJL19] show that the first derivative satisfies

$$2\,\mathrm{D}\,F(\Sigma) = I_d - \frac{1}{N}\sum_{i=1}^N \mathrm{GM}(\Sigma_i, \Sigma^{-1})\,.$$

We now compute the second derivative. Define the functions
$$\mathrm{inv}(\Sigma) := \Sigma^{-1},$$
$$\mathrm{conj}_A(\Sigma) := A\Sigma A,$$
$$\mathrm{sqrt}(\Sigma) := \Sigma^{1/2}.$$

For $Y \in \mathbb{S}^d$, the above maps have derivatives
$$\mathrm{D\,inv}(\Sigma)[Y] = -\Sigma^{-1}Y\Sigma^{-1},$$
$$\mathrm{D\,conj}_A(\Sigma)[Y] = AYA,$$
$$\mathrm{D\,sqrt}(\Sigma)[Y] = \int_0^\infty e^{-t\Sigma^{1/2}} Y e^{-t\Sigma^{1/2}}\,dt.$$

With these definitions in hand, we can write
$$2\,\mathrm{D}\,F(\Sigma) = I_d - \frac{1}{N}\sum_{i=1}^N \mathrm{conj}_{\Sigma_i^{1/2}} \circ \mathrm{sqrt} \circ \mathrm{conj}_{\Sigma_i^{-1/2}} \circ \mathrm{inv}(\Sigma).$$

Taking the derivative in a symmetric direction $Y \in \mathbb{S}^d$ and applying the chain rule repeatedly,

$2\,\mathrm{D}^2\,F(\Sigma)[Y]$

$$= \frac{1}{N}\sum_{i=1}^N \int_0^\infty \Sigma_i^{1/2} e^{-t\,(\Sigma_i^{1/2}\Sigma\Sigma_i^{1/2})^{-1/2}} \Sigma_i^{-1/2}\Sigma^{-1}Y\Sigma^{-1}\Sigma_i^{-1/2} e^{-t\,(\Sigma_i^{1/2}\Sigma\Sigma_i^{1/2})^{-1/2}} \Sigma_i^{1/2}\,dt.$$

Let $g(t,x) = \exp(-t/\sqrt{x})\,x^{-1}$ on $(t,x) \in (0,\infty)\times(0,\infty)$ and $Z_i = \Sigma_i^{1/2}\Sigma\Sigma_i^{1/2}$. Since $g(t,\cdot)$ is analytic on its domain, the Riesz–Dunford calculus (see [DS88]) applies and we may write

$$2\,\langle Y, \mathrm{D}^2\,F(\Sigma)[Y]\rangle_{\mathrm{F}} = \frac{1}{N}\sum_{i=1}^N \int_0^\infty \mathrm{tr}\big(g(t,Z_i)\Sigma_i^{1/2}Y\Sigma_i^{1/2} g(t,Z_i)\Sigma_i^{1/2}Y\Sigma_i^{1/2}\big)\,dt.$$

Using the spectral mapping theorem and Lemma 5 below we further write

$$\leq \frac{\|Y\|_{\mathrm{F}}^2}{N}\sum_{i=1}^N \lambda_{\max}(\Sigma_i)^2 \int_0^\infty \max_{\lambda \in \mathrm{spec}(Z_i)} g(t,\lambda)^2\,dt.$$

An analogous argument gives the lower bound

$$2\,\langle Y, \mathrm{D}^2\,F(\Sigma)[Y]\rangle_{\mathrm{F}} \geq \frac{\|Y\|_{\mathrm{F}}^2}{N}\sum_{i=1}^N \lambda_{\min}(\Sigma_i)^2 \int_0^\infty \min_{\lambda \in \mathrm{spec}(Z_i)} g(t,\lambda)^2\,dt.$$

To bound the integral, we note that

$$e^{-t/\sqrt{\lambda_{\min}(Z_i)}}\,\lambda_{\max}(Z_i)^{-1} \leq g(t,\lambda) \leq e^{-t/\sqrt{\lambda_{\max}(Z_i)}}\,\lambda_{\min}(Z_i)^{-1}$$

for all $\lambda \in \mathrm{spec}(Z_i)$. Since we assume $\alpha I_d \preceq \Sigma_i, \Sigma \preceq \beta I_d$, then $\alpha^2 I_d \preceq Z_i \preceq \beta^2 I_d$, so

$$2\,\langle Y, \mathrm{D}^2\,F(\Sigma)[Y]\rangle_{\mathrm{F}} \leq \beta^2 \int_0^\infty \exp\big(-\frac{2t}{\beta}\big)\frac{1}{\alpha^4}\,dt = \frac{\beta^3}{2\alpha^4}.$$

An analogous calculation for the lower bound finishes the proof. $\qquad\square$

*Remark* 4. Similar to Theorem 3, one can obtain improved strong convexity and smoothness parameters for $F$ based on non-uniform notions of conditioning.

We can now describe the projected GD and projected SGD updates. Let $\Pi_{\alpha,\beta} : \mathbb{S}^d \to \mathcal{K}_{\alpha,\beta}$ denote the Euclidean projection onto $\mathcal{K}_{\alpha,\beta}$ and let $\eta = 4\lambda_{\min}^4/\lambda_{\max}^3$. Given a starting matrix $\Sigma_0$, the projected GD scheme to minimize the barycenter functional of a measure $P$ supported on $\mathcal{K}_{\lambda_{\min},\lambda_{\max}}$ is given by
$$\Sigma_{t+1}^{\mathrm{EGD}} := \Pi_{\lambda_{\min},\lambda_{\max}}\big(\Sigma_t - \eta\,\mathrm{D}\,F(\Sigma_t^{\mathrm{EGD}})\big), \qquad t \geq 0. \tag{17}$$
Also, suppose that $\Sigma_1,\ldots,\Sigma_t$ are i.i.d. samples from $P$. Then, the projected stochastic gradient scheme is
$$\Sigma_{t+1}^{\mathrm{ESGD}} := \Pi_{\lambda_{\min},\lambda_{\max}}\big(\Sigma_t^{\mathrm{ESGD}} - \eta_{t+1}\{I_d - \Sigma_{t+1}\#(\Sigma_t^{\mathrm{ESGD}})^{-1}\}\big), \qquad t \geq 0, \tag{18}$$
where following [LSB12] we take the step size to be $\eta_t = 8\lambda_{\max}^4/(\lambda_{\min}^3\,(t+1))$.

We now state the convergence guarantees for these two algorithms.

**Theorem 9** (guarantees for Euclidean GD/SGD). *Assume that $P$ is supported on covariance matrices whose eigenvalues lie in the range $[\lambda_{\min}, \lambda_{\max}]$, $0 < \lambda_{\min} \leq \lambda_{\max} < \infty$. Let $\kappa := \lambda_{\max}/\lambda_{\min}$ denote the condition number. Assume that we initialize at $\Sigma_0 \in \operatorname{supp} P$.*

1. *(EGD) Let $\Sigma_T^{\mathrm{EGD}}$ denote the $T$-th iterate of projected Euclidean GD (17). Then,*

$$\|\Sigma_T^{\mathrm{EGD}} - \Sigma^\star\|_{\mathrm{F}}^2 \leq \exp\left(-\frac{T}{\kappa^7}\right) \|\Sigma_0 - \Sigma^\star\|_{\mathrm{F}}^2 . \tag{19}$$

2. *(ESGD) Let $\Sigma_T^{\mathrm{ESGD}}$ denote the $T$-th iterate of Euclidean projected SGD (18). Then,*

$$\mathbb{E}[\|\Sigma_T^{\mathrm{ESGD}} - \Sigma^\star\|_{\mathrm{F}}^2] \leq \frac{64 d \lambda_{\max}^2 \kappa^{6.5}}{T} .$$

*Proof.* (1) The preceding lemma shows that the barycenter functional $F$ is strongly convex and smooth with condition number $\kappa^7$. By [Bub15, Theorem 3.10], projected GD (17) converges at the stated rate.

(2) For ESGD, we must compute a bound on the Euclidean variance of the stochastic gradient. Using Lemma 2, we get the two-sided control

$$\frac{1}{\sqrt{\kappa}} I_d \preceq \Sigma_{t+1} \# (\Sigma_t^{\mathrm{ESGD}})^{-1} \preceq \sqrt{\kappa} \, I_d$$

and thus

$$\left\| I_d - \Sigma_{t+1} \# (\Sigma_t^{\mathrm{ESGD}})^{-1} \right\|_{\mathrm{F}}^2 \leq d \, (\sqrt{\kappa} - 1) \leq d \sqrt{\kappa} .$$

The result now follows from the preceding lemma and [LSB12]. $\qquad \square$

*Remark* 5. To compare the guarantees of Theorems 2 and 9, first we have

$$\frac{1}{2} \|\Sigma_T^{1/2} - \Sigma^{\star \, 1/2}\|_{\mathrm{F}}^2 \leq W_2^2(\Sigma_T, \Sigma^\star) \leq \|\Sigma_T^{1/2} - \Sigma^{\star \, 1/2}\|_{\mathrm{F}}^2$$

as a consequence of [CV21, Lemma 3.5]. Moreover, under our assumptions,

$$\frac{1}{4\lambda_{\max}} \|\Sigma_T - \Sigma^\star\|_{\mathrm{F}}^2 \leq \|\Sigma_T^{1/2} - \Sigma^{\star \, 1/2}\|_{\mathrm{F}}^2 \leq \frac{1}{4\lambda_{\min}} \|\Sigma_T - \Sigma^\star\|_{\mathrm{F}}^2 ,$$

where the first inequality is elementary and follows from

$$A - B = A^{1/2} \left( A^{1/2} - B^{1/2} \right) + \left( A^{1/2} - B^{1/2} \right) B^{1/2} ,$$

whereas the second inequality uses [Bha97, (X.46)].

For the iterations given by (17) and (18) to be practical, we need the projection step to be implementable. The following lemma takes care of this.

**Lemma 4.** *Let $\Pi_{\alpha,\beta} : \mathbb{S}^d \to \mathcal{K}_{\alpha,\beta}$ be the projection with respect to the Frobenius norm. Then*

$$\Pi_{\alpha,\beta}(Y) = \sum_{i=1}^d [(\lambda_i \wedge \beta) \vee \alpha] \, v_i v_i^{\mathsf{T}}$$

*where $Y = \sum_{i=1}^d \lambda_i v_i v_i^{\mathsf{T}}$ is an orthogonal eigendecomposition of $Y$.*

*Proof.* Let $Y = Q \Lambda Q^{\mathsf{T}}$ be an orthogonal eigendecomposition of $Y$. Since the Frobenius norm is unitarily invariant, we have

$$\Pi_{\alpha,\beta}(Y) = \operatorname*{arg\,min}_{X \in \mathcal{K}_{\alpha,\beta}} \|X - Q\Lambda Q^{\mathsf{T}}\|_{\mathrm{F}}^2 = \operatorname*{arg\,min}_{X \in \mathcal{K}_{\alpha,\beta}} \|Q^{\mathsf{T}} X Q - \Lambda\|_{\mathrm{F}}^2 = Q \left( \operatorname*{arg\,min}_{X \in \mathcal{K}_{\alpha,\beta}} \|X - \Lambda\|_{\mathrm{F}}^2 \right) Q^{\mathsf{T}}$$

and the result follows. $\qquad \square$

Finally, we state and prove the elementary lemma we used in the proof of Lemma 3.

**Lemma 5.** *Let $A, B \in \mathbb{S}_{++}^d$ and $Y \in \mathbb{S}^d$. Then*

$$\lambda_{\min}(A) \, \lambda_{\min}(B) \, \|Y\|_{\mathrm{F}}^2 \leq \operatorname{tr}(AYBY) \leq \lambda_{\max}(A) \, \lambda_{\max}(B) \, \|Y\|_{\mathrm{F}}^2 .$$

*Proof.* The result follows immediately from $\operatorname{tr}(AYBY) = \|A^{1/2} Y B^{1/2}\|_{\mathrm{F}}^2$ and $\lambda_{\min}(A^{1/2}) = \lambda_{\min}(A)^{1/2}$ (similarly for $B$). $\qquad \square$

## C.3 SDP formulation

The SDP formulation of the Bures-Wasserstein barycenter is as follows. Suppose that $P$ is a discrete distribution, $P = \sum_{i=1}^{k} p_i \delta_{\Sigma_i}$. The Wasserstein distance between $\Sigma_0, \Sigma_1 \in \mathbb{S}_{++}^d$ can be expressed as

$$W_2^2(\Sigma_0, \Sigma_1) = \min_{S \in \mathbb{R}^{d \times d}} \left\{ \text{tr}(\Sigma_0 + \Sigma_1 - 2S) \quad \text{such that} \quad \begin{bmatrix} \Sigma_0 & S \\ S^\mathsf{T} & \Sigma_1 \end{bmatrix} \succeq 0 \right\}.$$

It follows that the barycenter $\Sigma^\star$ of $P$ solves the optimization problem

$$\min_{\substack{\Sigma^\star \in \mathbb{S}_{++}^d \\ S_1, \dots, S_k \in \mathbb{R}^{d \times d}}} \left\{ \text{tr}\left(\Sigma^\star - 2 \sum_{i=1}^{k} p_i S_i\right) \quad \text{such that} \quad \begin{bmatrix} \Sigma_i & S_i \\ S_i^\mathsf{T} & \Sigma^\star \end{bmatrix} \succeq 0, \ \forall i \in [k] \right\}.$$

## D   Proofs for entropically-regularized barycenters

We begin by remarking how the non-centered case can be reduced to the centered case.

*Remark 6.* For a probability measure $\mu$, let $m_\mu$ denote its mean and let $\bar{\mu}$ denote the centered version of $\mu$. Using Fact 5 in Appendix A.5, one can verify that

$$\frac{1}{2} \int W_2^2(b, \mu) \, dP(\mu) + \gamma \, \text{KL}\big(b \, \big\| \, \mathcal{N}(0, I_d)\big)$$

$$= \frac{1}{2} \int \|m_b - m_\mu\|^2 \, dP(\mu) + \frac{\gamma}{2} \|m_b\|^2 + \frac{1}{2} \int W_2^2(\bar{b}, \bar{\mu}) \, dP(\mu) + \gamma \, \text{KL}\big(\bar{b} \, \big\| \, \mathcal{N}(0, I_d)\big).$$

This shows that the objective of the entropically-regularized barycenter decouples into two parts, one involving the mean of $b$ and the other involving the centered version of $b$. Explicitly, we can compute

$$m^\star := \frac{1}{1 + \gamma} \int m_\mu \, dP(\mu)$$

and the entropically-regularized barycenter $\bar{b}^\star$ of the centered versions of the distributions in $P$. Then, if $\tau : \mathbb{R}^d \to \mathbb{R}^d$ denotes the translation $x \mapsto x + m^\star$, the solution to the original entropically-regularized barycenter problem is $\tau_\# \bar{b}^\star$.

We now overview the proof strategy; proofs are then provided in the subsequent subsections. Throughout this section let $P$ be supported on $\mathcal{K}_{1/\sqrt{\kappa}, \sqrt{\kappa}}$, the subset of matrices in $\mathbb{S}_{++}^d$ with eigenvalues in the range $[1/\sqrt{\kappa}, \sqrt{\kappa}]$.

An important observation driving our analysis is that the gradient of the KL divergence at $\Sigma$ has the following form:

$$\nabla \, \text{KL}(\cdot \, \| \, I_d)(\Sigma) = I_d - \Sigma^{-1} = I_d - T_{\Sigma \to \Sigma^{-1}} = -\log_\Sigma(\Sigma^{-1}). \tag{20}$$

This can be shown by observing that

$$\text{KL}(\Sigma \, \| \, I_d) = \frac{1}{2} \, \text{tr} \, \Sigma - \frac{1}{2} \ln \det \Sigma - \frac{d}{2},$$

computing the Euclidean gradient, and appealing to Fact 1 in Appendix A.5. This gradient identity is convenient for applying our generalized geodesic convexity results and allows us to prove the following Lemma in Subsection D.1. Put $\Sigma^+ := \exp_\Sigma(-\eta \nabla F_\gamma(\Sigma))$.

**Lemma 6.** *If $\Sigma \in \mathcal{K}_{1/\sqrt{\kappa}, \sqrt{\kappa}}$, then so is $\Sigma^+$.*

We also establish a couple of properties of our objective function in Subsection D.2.

**Proposition 4.** *Define $G : \mathcal{K}_{1/\sqrt{\kappa}, \sqrt{\kappa}} \to \mathbb{R}$ to take $\Sigma \mapsto \text{KL}(\Sigma \, \| \, I_d)$. Then, the following hold:*

1. *$G$ is $2\sqrt{\kappa}$-smooth with respect to Wasserstein geodesics.*

2. *$F_\gamma$ is $1/(4\kappa^{7/2})$-strongly convex with respect to Euclidean geodesics on $\mathcal{K}_{1/\sqrt{\kappa}, \sqrt{\kappa}}$.*

3. *$F_\gamma$ is strictly convex on all of $\mathbb{S}_{++}^d$.*

With these facts, we can establish existence and uniqueness of $\Sigma^\star$ and prove Proposition 1 in Subsection D.3.

Next we prove smoothness and PL inequalities in Subsection D.4.

**Lemma 7** (Smoothness). *If $\Sigma \in \mathcal{K}_{1/\sqrt{\kappa},\sqrt{\kappa}}$ and we take the step size $\eta = 1/(1 + 2\gamma\sqrt{\kappa})$, then*

$$F_\gamma(\Sigma^+) - F_\gamma(\Sigma) \leq -\frac{1}{2(1 + 2\gamma\sqrt{\kappa})} \|\nabla F_\gamma(\Sigma)\|_\Sigma^2 .$$

**Lemma 8** (PL inequality). *If $\Sigma \in \mathcal{K}_{1/\sqrt{\kappa},\sqrt{\kappa}}$, then*

$$F_\gamma(\Sigma) - F_\gamma(\Sigma^\star) \leq \frac{\kappa^4}{2} \|\nabla F_\gamma(\Sigma)\|_\Sigma^2 .$$

When the regularization parameter $\gamma$ is large, we can instead use a different argument to improve the PL constant.

**Lemma 9** (PL inequality, large regularization). *If $\Sigma \in \mathcal{K}_{1/\sqrt{\kappa},\sqrt{\kappa}}$ and $\gamma \geq 14\kappa^4$, then*

$$F_\gamma(\Sigma) - F_\gamma(\Sigma^\star) \leq \frac{1}{\gamma} \|\nabla F_\gamma(\Sigma)\|_\Sigma^2 .$$

The main theorem now follows by combining these lemmas.

*Proof of Theorem 4.* By Lemma 6, the Lemmas 7, 8, and 9 hold throughout the optimization trajectory. Let $C = 2/\kappa^4$ if we apply Lemma 8, and let $C = \gamma$ if we apply Lemma 9. Then,

$$\begin{aligned}
F_\gamma(\Sigma_{t+1}) - F_\gamma(\Sigma^\star) &= F_\gamma(\Sigma_{t+1}) - F_\gamma(\Sigma_t) + F_\gamma(\Sigma_t) - F_\gamma(\Sigma^\star) \\
&\leq -\frac{1}{2(1 + 2\gamma\sqrt{\kappa})} \|\nabla F_\gamma(\Sigma_t)\|_{\Sigma_t}^2 + F_\gamma(\Sigma_t) - F_\gamma(\Sigma^\star) \\
&\leq \left(1 - \frac{C}{2(1 + 2\gamma\sqrt{\kappa})}\right) \{F_\gamma(\Sigma_t) - F_\gamma(\Sigma^\star)\} .
\end{aligned}$$

Iterating yields the result. $\qquad\square$

## D.1 Trapping the iterates

*Proof of Lemma 6.* Combining (20) with the formula for the gradient of the squared Bures-Wasserstein distance (Fact 2 in Appendix A.5), we see that in fact

$$-\nabla F_\gamma(\Sigma) = \int \log_\Sigma(\Sigma') \, dP(\Sigma') + \gamma \log_\Sigma(\Sigma^{-1}) .$$

We then apply Theorem 1 (see also the discussion in Appendix A.2) and the fact that that $\eta(1+\gamma) \leq 1$ to yield

$$\begin{aligned}
\sqrt{\lambda_{\min}}(\Sigma^+) &= \sqrt{\lambda_{\min}} \circ \exp_\Sigma\left(\eta \int \log_\Sigma(\Sigma') \, dP(\Sigma') + \eta\gamma \log_\Sigma(\Sigma^{-1}) + (1 - \eta - \eta\gamma)\underbrace{\log_\Sigma(\Sigma)}_{=0}\right) \\
&\geq \eta \int \sqrt{\lambda_{\min}}(\Sigma') \, dP(\Sigma') + \eta\gamma\sqrt{\lambda_{\min}}(\Sigma^{-1}) + [1 - \eta(1+\gamma)] \sqrt{\lambda_{\min}}(\Sigma) \\
&\geq \frac{1}{\kappa^{1/4}} .
\end{aligned}$$

The analogous argument shows that $\sqrt{\lambda_{\max}}(\Sigma^+) \leq \kappa^{1/4}$, hence the result. $\qquad\square$

## D.2 Properties of the KL divergence

*Proof of Proposition 4.* For the first claim, fix $\Sigma_0, \Sigma_1 \in \mathcal{K}_{1/\sqrt{\kappa},\sqrt{\kappa}}$ and let $T$ denote the transport map from $\Sigma_0$ to $\Sigma_1$. Then put

$$\Sigma_s := \big((1-s)I_d + sT\big)\Sigma_0\big((1-s)I_d + sT\big) = (1-s)^2 \Sigma_0 + s^2 \Sigma_1 + s(1-s)(\Sigma_0 T + T\Sigma_0) .$$

In other words, $(\Sigma_s)_{s\in[0,1]}$ is the Bures-Wasserstein geodesic between $\Sigma_0$ and $\Sigma_1$ (see (10)). It suffices to show (see Appendix A.3) that

$$\partial_s^2 \, \mathrm{KL}(\Sigma_s \parallel I_d)|_{s=0} \leq 2\sqrt{\kappa} \, W_2^2(\Sigma_0, \Sigma_1).$$

Since $\mathrm{KL}(\Sigma_s \parallel I_d) = \frac{1}{2}(\mathrm{tr}(\Sigma_s) - \ln\det\Sigma_s + \text{constant})$, we analyze the first two terms separately. First, we note that

$$\partial_s^2 \, \mathrm{tr}(\Sigma_s)|_{s=0} = 2\,\mathrm{tr}(\Sigma_0 + \Sigma_1 - 2\Sigma_0 T) = 2W_2^2(\Sigma_0, \Sigma_1),$$

where the equality follows from (9). For the second term we start by observing that

$$-\ln\det\Sigma_s = -\ln\det\Sigma_0 - 2\ln\det\big((1-s)I_d + sT\big).$$

Using this identity we see that

$$-\partial_s^2\Big(\frac{1}{2}\ln\det(\Sigma_s)\Big)\Big|_{s=0} = \mathrm{tr}\big((T - I_d)^2\big) \leq \sqrt{\kappa}\,\|T - I_d\|_{\Sigma_0}^2 = \sqrt{\kappa}\,W_2^2(\Sigma_0, \Sigma_1).$$

Putting these bounds together yields the result.

For the second claim, using the convexity of $-\ln\det$, it follows that the Euclidean Hessian of $F_\gamma$ satisfies $\mathrm{D}^2\,F_\gamma \succeq \mathrm{D}^2\,F$, where $F$ is the barycenter functional. It follows from Lemma 3 that

$$\mathrm{D}^2\,F_\gamma \succeq \frac{1}{4\kappa^{7/2}}\,I_d$$

on the set $\mathcal{K}_{1/\sqrt{\kappa},\sqrt{\kappa}}$.

Finally, for the third claim we can use the convexity of $F$ and the strict convexity of $-\ln\det$ (together with $\gamma > 0$) to argue that $\mathrm{D}^2\,F_\gamma \succ 0$ on $\mathbb{S}_{++}^d$. $\qquad\square$

### D.3  Existence and uniqueness of the minimizer

*Proof of Proposition 1.* First, we prove that when restricted to $\mathbb{S}_{++}^d$, the functional $F_\gamma$ has a unique minimizer. Let $H \colon \mathcal{K}_{1/\sqrt{\kappa},\sqrt{\kappa}} \to \mathbb{S}_{++}^d$ take

$$\Sigma \mapsto \exp_\Sigma\Big(-\frac{1}{1+\gamma}\,\nabla F_\gamma(\Sigma)\Big).$$

Then by an analogous calculation to the proof of Lemma 6, $H$ must map into $\mathcal{K}_{1/\sqrt{\kappa},\sqrt{\kappa}}$. We may thus apply Brouwer's fixed point theorem to guarantee a fixed point of $H$ in $\mathcal{K}_{1/\sqrt{\kappa},\sqrt{\kappa}}$, call it $\Sigma^\star$. Note that this means precisely that $\nabla F_\gamma(\Sigma^\star) = 0$. By the equivalence of Euclidean and Bures-Wasserstein gradients (Fact 1 in Appendix A.5), we conclude that $\mathrm{D}\,F_\gamma(\Sigma^\star) = 0$ as well. By the strict convexity of $F_\gamma$ (the third claim of Proposition 4), we deduce that $\Sigma^\star$ is the unique minimizer of $F_\gamma$ on $\mathbb{S}_{++}^d$ (actually, on all of $\mathbb{S}_+^d$, since $-\ln\det$ blows up if the determinant approaches 0).

Next, let $b$ be a probability measure on $\mathbb{R}^d$ which has mean $m$ and covariance matrix $\Sigma$. Let $\bar{b}$ denote the centered version of $b$. We now claim that

$$F_\gamma(b) \geq F_\gamma(\bar{b}) \geq F_\gamma\big(\mathcal{N}(0,\Sigma)\big) \geq F_\gamma\big(\mathcal{N}(0,\Sigma^\star)\big).$$

The first inequality is due to Remark 6 and it is strict unless $b = \bar{b}$. The second inequality follows from Fact 6 in Appendix A.5, together with the classical fact that the Gaussian maximizes entropy among all centered distributions with the same covariance matrix; this latter fact is proven in [CT06, Theorem 8.6.5], and it also shows that the inequality is strict unless $\bar{b} = \mathcal{N}(0,\Sigma)$. Finally, the last inequality is what we have shown above, and it is also strict unless $\Sigma = \Sigma^\star$. $\qquad\square$

### D.4  PL and smoothness inequalities

*Proof of Lemma 7.* By Proposition 4 and the 1-geodesic smoothness of the barycenter functional [Che+20, Theorem 7] we deduce that $F_\gamma = F + \gamma G$ is $(1 + 2\gamma\sqrt{\kappa})$-smooth, i.e.

$$F_\gamma(\Sigma^+) - F_\gamma(\Sigma) \leq \langle \nabla F_\gamma(\Sigma), \log_\Sigma(\Sigma^+)\rangle_\Sigma + \frac{1 + 2\gamma\sqrt{\kappa}}{2}\,W_2^2(\Sigma, \Sigma^+).$$

Substituting in $\log_\Sigma(\Sigma^+) = -\eta\nabla F_\gamma(\Sigma)$ and the step size $\eta = 1/(1 + 2\gamma\sqrt{\kappa})$ yields the result. $\quad\square$

*Proof of Lemma 8.* From the second claim in Proposition 4, and since $\mathcal{K}_{1/\sqrt{\kappa},\sqrt{\kappa}}$ is convex with respect to Euclidean geodesics, we see that for $\Sigma \in \mathcal{K}_{1/\sqrt{\kappa},\sqrt{\kappa}}$

$$F_\gamma(\Sigma) - F_\gamma(\Sigma^\star) \leq \langle \mathrm{D}\, F_\gamma(\Sigma), \Sigma - \Sigma^\star \rangle - \frac{1}{8\kappa^{7/2}} \|\Sigma - \Sigma^\star\|_{\mathrm{F}}^2$$

$$= \frac{1}{2} \langle \nabla F_\gamma(\Sigma), \Sigma - \Sigma^\star \rangle - \frac{1}{8\kappa^{7/2}} \|\Sigma - \Sigma^\star\|_{\mathrm{F}}^2,$$

where the last line uses Fact 1 in Appendix A.5. Next we observe that by combining Cauchy-Schwarz with Young's inequality we get that for all $r > 0$,

$$\frac{1}{2} \langle \nabla F_\gamma(\Sigma), \Sigma - \Sigma^\star \rangle \leq \frac{1}{2} \|\nabla F_\gamma(\Sigma)\|_\Sigma \|\Sigma - \Sigma^\star\|_{\Sigma^{-1}} \leq \frac{r}{16} \|\nabla F_\gamma(\Sigma)\|_\Sigma^2 + \frac{1}{r} \|\Sigma - \Sigma^\star\|_{\Sigma^{-1}}^2$$

$$\leq \frac{r}{16} \|\nabla F_\gamma(\Sigma)\|_\Sigma^2 + \frac{\sqrt{\kappa}}{r} \|\Sigma - \Sigma^\star\|_{\mathrm{F}}^2.$$

Putting $r = 8\kappa^4$ yields the result. $\qquad\square$

*Proof of Lemma 9.* From the 1-smoothness of the barycenter functional $F$ [Che+20, Theorem 7],

$$F(\Sigma) - F(\Sigma^\star) \leq \langle \nabla F(\Sigma^\star), T_{\Sigma^\star \to \Sigma} - I_d \rangle_{\Sigma^\star} + \frac{1}{2} W_2^2(\Sigma, \Sigma^\star)$$

$$= -\langle \nabla F(\Sigma^\star) T_{\Sigma \to \Sigma^\star}, T_{\Sigma \to \Sigma^\star} - I_d \rangle_\Sigma + \frac{1}{2} W_2^2(\Sigma, \Sigma^\star).$$

On the other hand, it is a celebrated fact that $G$ is 1-strongly convex w.r.t. the Wasserstein geometry (see [Vil03, §5]; on the Bures-Wasserstein space, it can also be read off from the proof of Proposition 4). It implies

$$G(\Sigma) - G(\Sigma^\star) \leq -\langle \nabla G(\Sigma), T_{\Sigma \to \Sigma^\star} - I_d \rangle_\Sigma - \frac{1}{2} W_2^2(\Sigma, \Sigma^\star).$$

Combining these inequalities yields

$$F_\gamma(\Sigma) - F_\gamma(\Sigma^\star) \leq -\langle \nabla F_\gamma(\Sigma), T_{\Sigma \to \Sigma^\star} - I_d \rangle_\Sigma$$

$$+ \langle \nabla F(\Sigma) - \nabla F(\Sigma^\star) T_{\Sigma \to \Sigma^\star}, T_{\Sigma \to \Sigma^\star} - I_d \rangle_\Sigma - \frac{\gamma - 1}{2} W_2^2(\Sigma, \Sigma^\star).$$

We next bound $\|\nabla F(\Sigma) - \nabla F(\Sigma^\star) T_{\Sigma \to \Sigma^\star}\|_\Sigma$. Write

$$\|\nabla F(\Sigma) - \nabla F(\Sigma^\star) T_{\Sigma \to \Sigma^\star}\|_\Sigma = \left\| \int \{T_{\Sigma \to \Sigma'} - T_{\Sigma^\star \to \Sigma'} T_{\Sigma \to \Sigma^\star} + T_{\Sigma \to \Sigma^\star} - I_d\} \,\mathrm{d}P(\Sigma') \right\|_\Sigma$$

$$\leq \left\| \int \{T_{\Sigma \to \Sigma'} - T_{\Sigma^\star \to \Sigma'} T_{\Sigma \to \Sigma^\star}\} \,\mathrm{d}P(\Sigma') \right\|_\Sigma + W_2(\Sigma, \Sigma^\star).$$

For the first term, start with the triangle inequality,

$$\|T_{\Sigma \to \Sigma'} - T_{\Sigma^\star \to \Sigma'} T_{\Sigma \to \Sigma^\star}\|_\Sigma \leq \|T_{\Sigma \to \Sigma'} - T_{\Sigma^\star \to \Sigma'} + T_{\Sigma^\star \to \Sigma'} - T_{\Sigma^\star \to \Sigma'} T_{\Sigma \to \Sigma^\star}\|_\Sigma$$

$$\leq \|T_{\Sigma \to \Sigma'} - T_{\Sigma^\star \to \Sigma'}\|_\Sigma + \|T_{\Sigma^\star \to \Sigma'} - T_{\Sigma^\star \to \Sigma'} T_{\Sigma \to \Sigma^\star}\|_\Sigma$$

$$\leq \|T_{\Sigma \to \Sigma'} - T_{\Sigma^\star \to \Sigma'}\|_\Sigma + \sqrt{\kappa} \|I_d - T_{\Sigma \to \Sigma^\star}\|_\Sigma$$

$$= \|T_{\Sigma \to \Sigma'} - T_{\Sigma^\star \to \Sigma'}\|_\Sigma + \sqrt{\kappa}\, W_2(\Sigma, \Sigma^\star),$$

where we use the fact that the eigenvalues of the transport map $T_{\Sigma^\star \to \Sigma'}$ are bounded in magnitude by $\sqrt{\kappa}$ (Lemma 2). Next, consider the distribution $\delta_{\Sigma'}$ (a point mass on $\Sigma'$) and let $F^{\Sigma'}$ denote the barycenter functional for the distribution $\delta_{\Sigma'}$. Then, the Euclidean smoothness bound for $F^{\Sigma'}$ (Lemma 3) yields

$$\|\mathrm{D}\, F^{\Sigma'}(\Sigma) - \mathrm{D}\, F^{\Sigma'}(\Sigma^\star)\|_{\mathrm{F}} \leq \frac{\kappa^{7/2}}{4} \|\Sigma - \Sigma^\star\|_{\mathrm{F}}.$$

Hence,

$$\|T_{\Sigma \to \Sigma'} - T_{\Sigma^\star \to \Sigma'}\|_\Sigma^2 = \|\mathrm{D}\, F^{\Sigma'}(\Sigma) - \mathrm{D}\, F^{\Sigma'}(\Sigma^\star)\|_\Sigma^2 \leq \sqrt{\kappa} \|\mathrm{D}\, F^{\Sigma'}(\Sigma) - \mathrm{D}\, F^{\Sigma'}(\Sigma^\star)\|_{\mathrm{F}}^2$$

$$\leq \frac{\kappa^{15/2}}{16} \|\Sigma - \Sigma^\star\|_{\mathrm{F}}^2 \leq \frac{\kappa^8}{2} W_2^2(\Sigma, \Sigma^\star),$$

where the last inequality follows from Remark 5. Putting these inequalities together,

$$\|\nabla F(\Sigma) - \nabla F(\Sigma^\star)T_{\Sigma \to \Sigma^\star}\|_\Sigma \leq 3\kappa^4 \, W_2(\Sigma, \Sigma^\star) \,.$$

Continuing from before, we obtain

$$
\begin{aligned}
F_\gamma(\Sigma) - F_\gamma(\Sigma^\star) &\leq \|\nabla F_\gamma(\Sigma)\|_\Sigma \, W_2(\Sigma, \Sigma^\star) \\
&\quad + \|\nabla F(\Sigma) - \nabla F(\Sigma^\star)T_{\Sigma \to \Sigma^\star}\|_\Sigma \, W_2(\Sigma, \Sigma^\star) - \frac{\gamma - 1}{2} \, W_2^2(\Sigma, \Sigma^\star) \\
&\leq \|\nabla F_\gamma(\Sigma)\|_\Sigma \, W_2(\Sigma, \Sigma^\star) + 3\kappa^4 \, W_2^2(\Sigma, \Sigma^\star) - \frac{\gamma - 1}{2} \, W_2^2(\Sigma, \Sigma^\star) \\
&\leq \|\nabla F_\gamma(\Sigma)\|_\Sigma \, W_2(\Sigma, \Sigma^\star) - \frac{\gamma}{4} \, W_2^2(\Sigma, \Sigma^\star)
\end{aligned}
$$

provided that $\gamma$ is sufficiently large, $\gamma \geq 14\kappa^4$. For this large regularization, we can then prove

$$
\begin{aligned}
F_\gamma(\Sigma) - F_\gamma(\Sigma^\star) &\leq \frac{1}{\gamma} \, \|\nabla F_\gamma(\Sigma)\|_\Sigma^2 + \frac{\gamma}{4} \, W_2^2(\Sigma, \Sigma^\star) - \frac{\gamma}{4} \, W_2^2(\Sigma, \Sigma^\star) \\
&\leq \frac{1}{\gamma} \, \|\nabla F_\gamma(\Sigma)\|_\Sigma^2 \,.
\end{aligned}
$$

This completes the proof. $\qquad\square$

# E    Proofs for geometric medians

## E.1    Convergence guarantee for smoothed Riemannian gradient descent

We begin with the proof of Proposition 2.

*Proof of Proposition 2.* Let $F : \mathcal{P}_2(\mathbb{R}^d) \to \mathbb{R}$ be the geometric median functional, $F(b) := \int W_2(b, \cdot) \, dP$. If we regard $F$ as a functional over the Bures-Wasserstein space, then by continuity of $F$ and compactness of the set $\{\|\cdot\| \leq \lambda_{\max}\} \subseteq \mathbb{S}_+^d$, there exists a minimizer $\Sigma^\star_{\mathrm{median}}$ of $F$ on this set. We will show that the Gaussian $b^\star_{\mathrm{median}}$ with covariance $\Sigma^\star_{\mathrm{median}}$ minimizes $F$ over all of Wasserstein space.

First, recall the map $\mathrm{clip}^{\lambda_{\max}}$ in Proposition 3, which is a contraction w.r.t. the Bures-Wasserstein metric. Then, for any $\Sigma \in \mathbb{S}_+^d$, it holds that

$$
\begin{aligned}
F(\Sigma) = \int W_2(\Sigma, \Sigma') \, dP(\Sigma') &\geq \int W_2(\mathrm{clip}^{\lambda_{\max}} \Sigma, \Sigma') \, dP(\Sigma') \\
&\geq \int W_2(\Sigma^\star_{\mathrm{median}}, \Sigma') \, dP(\Sigma') = F(\Sigma^\star_{\mathrm{median}}) \,,
\end{aligned}
$$

so that $\Sigma^\star_{\mathrm{median}}$ minimizes $F$ over $\mathbb{S}_+^d$.

Next, using Fact 6 in Appendix A.5, if $b \in \mathcal{P}_2(\mathbb{R}^d)$ has covariance matrix $\Sigma$, then

$$F(b) = \int W_2(b, \cdot) \, dP \geq \int W_2(\gamma_{0,\Sigma}, \cdot) \, dP \geq \int W_2(\Sigma^\star_{\mathrm{median}}, \cdot) \, dP = F(b^\star_{\mathrm{median}}) \,,$$

so that $b^\star_{\mathrm{median}}$ minimizes $F$ over $\mathcal{P}_2(\mathbb{R}^d)$.

By definition, $\Sigma^\star_{\mathrm{median}}$ has eigenvalues upper bounded by $\lambda_{\max}$. To finish the proof, we must show that the eigenvalues are also lower bounded by $\lambda_{\min}$; we defer this part of the proof until the end of this section. $\qquad\square$

As the main difficulty in the analysis of the geometric median is the lack of both convexity and smoothness, we now pause to justify these remarks.

*Remark 7.* We claim that the unsquared Wasserstein distance $W_2(\cdot, \Sigma')$ is neither geodesically convex nor geodesically smooth. For the former statement, note that the geodesic convexity of $W_2(\cdot, \Sigma')$ would imply the geodesic convexity of $W_2^2(\cdot, \Sigma')$, but the squared Wasserstein distance is known

to not be geodesically convex (in general, it is not even *semi-convex*, see [AGS08, Example 9.1.5]; for a Gaussian example, see [Che+20, Appendix B.2]). In fact, unsquared metrics are almost never geodesically smooth; if this were the case, then there would exist a constant $\beta < \infty$ for which $W_2(\Sigma, \Sigma') \leq \frac{\beta}{2} W_2^2(\Sigma, \Sigma')$, which is manifestly false.

Moreover, the function $W_2(\cdot, \Sigma')$ is neither Euclidean convex nor Euclidean smooth. To see this, observe that in one dimension we have $W_2(\Sigma, \Sigma') = |\sqrt{\Sigma} - \sqrt{\Sigma'}|$ (see Fact 4 in Appendix A.5), which is neither convex nor smooth. It is notable that for this one-dimensional example, $W_2(\Sigma, \Sigma')$ is convex with respect to the variable $\sqrt{\Sigma}$, but it appears that reparameterization does not help in general; numerics indicate that the function $A \mapsto W_2(A^2, \Sigma')$ is not Euclidean convex on $\mathbb{S}_{++}^d$.

We now proceed with the analysis of the smoothed Riemannian GD algorithm given as Algorithm 4. Recall that $F_\varepsilon$ denotes the smoothed geometric median functional. The first step is to show that the smoothing does not affect the objective significantly.

**Lemma 10.** *For any $\Sigma \in \mathbb{S}_{++}^d$, we have $|F(\Sigma) - F_\varepsilon(\Sigma)| \leq \varepsilon$.*

*Proof.* This follows from

$$\left|W_2(\Sigma, \Sigma') - \sqrt{W_2^2(\Sigma, \Sigma') + \varepsilon^2}\right| = \left|\sqrt{W_2^2(\Sigma, \Sigma')} - \sqrt{W_2^2(\Sigma, \Sigma') + \varepsilon^2}\right| \leq \varepsilon$$

and integrating. $\qquad\square$

Hence, if we can find a point $\hat{\Sigma}$ with $F_\varepsilon(\hat{\Sigma}) - \inf F_\varepsilon \leq \varepsilon$, it will then follow that $F(\hat{\Sigma}) - \inf F \leq 3\varepsilon$.

We next show that replacing $W_2$ by $W_{2,\varepsilon}$ indeed yields smoothness.

**Lemma 11.** *The functional $F_\varepsilon$ is $1/\varepsilon$-geodesically smooth.*

*Proof.* Recall from Theorem 7 that one-half of the squared Wasserstein distance is 1-smooth. This means that for any $W_2$ geodesic $(\Sigma_t)_{t\in\mathbb{R}}$, the following Hessian bound holds:

$$\frac{1}{2} \partial_t^2|_{t=0} W_2^2(\Sigma_t, \Sigma') \leq \|\dot{\Sigma}_0\|_{\Sigma_0}^2.$$

Here, $\dot{\Sigma}_0$ denotes the Bures-Wasserstein tangent vector, see the end of Appendix A.1. We use this to compute the smoothness of $F_\varepsilon$. Riemannian calculus yields

$$\partial_t F_\varepsilon(\Sigma_t) = \int \frac{\partial_t W_2^2(\Sigma_t, \Sigma')}{2 W_{2,\varepsilon}(\Sigma_t, \Sigma')} \, \mathrm{d}P(\Sigma'),$$

$$\partial_t^2|_{t=0} F_\varepsilon(\Sigma_t) = \int \left[\frac{\partial_t^2|_{t=0} W_2^2(\Sigma_t, \Sigma')}{2 W_{2,\varepsilon}(\Sigma_0, \Sigma')} - \frac{\{\partial_t|_{t=0} W_2^2(\Sigma_t, \Sigma')\}^2}{4 W_{2,\varepsilon}^3(\Sigma_0, \Sigma')}\right] \, \mathrm{d}P(\Sigma').$$

The second term is non-positive. For the first term,

$$\int \frac{\partial_t^2|_{t=0} W_2^2(\Sigma_t, \Sigma')}{2 W_{2,\varepsilon}(\Sigma_0, \Sigma')} \, \mathrm{d}P(\Sigma') = \int \underbrace{\frac{1}{W_{2,\varepsilon}(\Sigma_0, \Sigma')}}_{\leq 1/\varepsilon} \underbrace{\frac{1}{2} \partial_t^2|_{t=0} W_2^2(\Sigma_t, \Sigma')}_{\leq \|\dot{\Sigma}_0\|_{\Sigma_0}^2} \, \mathrm{d}P(\Sigma') \leq \frac{1}{\varepsilon} \|\dot{\Sigma}_0\|_{\Sigma_0}^2.$$

Hence, $F_\varepsilon$ is $1/\varepsilon$-smooth. $\qquad\square$

The next step is to prove a gradient domination condition for the functional $F_\varepsilon$. Let $\Sigma_\varepsilon^\star$ denote a minimizer of $F_\varepsilon$.

**Lemma 12.** *Suppose that the covariance matrices in the support of $P$, as well as $\Sigma$ itself, have eigenvalues which lie in $[\lambda_{\min}, \lambda_{\max}]$. Let $\kappa := \lambda_{\max}/\lambda_{\min}$ denote the condition number. Then,*

$$F_\varepsilon(\Sigma) - F_\varepsilon(\Sigma_\varepsilon^\star) \leq 2\kappa^{1/4} F_\varepsilon(\Sigma) \|\nabla F_\varepsilon(\Sigma)\|_\Sigma^{1/2}.$$

*Proof.* As the first part of the proof relies on general optimal transport arguments, we use the notation of general Wasserstein space. Given two measures $\mu, \nu$, let $\varphi_{\mu\to\nu}$ denote the Kantorovich potential

from $\mu$ to $\nu$, and also denote $\psi_{\mu\to\nu} := \|\cdot\|^2/2 - \varphi_{\mu\to\nu}$. Also, let $b_\varepsilon^\star$ denote the minimizer of $F_\varepsilon$. Then, from Kantorovich duality,

$$W_2(b, \mu) = \sqrt{2\int \psi_{\mu\to b}\,\mathrm{d}\mu + 2\int \psi_{b\to\mu}\,\mathrm{d}b}\,,$$

$$W_2(b_\varepsilon^\star, \mu) \geq \sqrt{\max\Big\{2\int \psi_{\mu\to b}\,\mathrm{d}\mu + 2\int \psi_{b\to\mu}\,\mathrm{d}b_\varepsilon^\star, 0\Big\}}\,.$$

It follows that

$$W_{2,\varepsilon}(b, \mu) - W_{2,\varepsilon}(b_\varepsilon^\star, \mu) \leq \sqrt{2\max\Big\{\int \psi_{b\to\mu}\,\mathrm{d}(b - b_\varepsilon^\star), 0\Big\}}\,.$$

Integrating,

$$F_\varepsilon(b) - F_\varepsilon(b_\varepsilon^\star) \leq \int \sqrt{2\,\Big|\int \psi_{b\to\mu}\,\mathrm{d}(b - b_\varepsilon^\star)\Big|}\,\mathrm{d}P(\mu)$$

$$= \int \sqrt{2W_{2,\varepsilon}(b, \mu)\,\Big|\int \frac{\psi_{b\to\mu}}{W_{2,\varepsilon}(b, \mu)}\,\mathrm{d}(b - b_\varepsilon^\star)\Big|}\,\mathrm{d}P(\mu)$$

$$\leq \sqrt{2\int W_{2,\varepsilon}(b, \mu)\,\mathrm{d}P(\mu)\,\Big|\iint \frac{\psi_{b\to\mu}}{W_{2,\varepsilon}(b, \mu)}\,\mathrm{d}(b - b_\varepsilon^\star)\,\mathrm{d}P(\mu)\Big|}\,.$$

Following [Che+20], we introduce the constant-speed $W_2$ geodesic $(b_s)_{s\in[0,1]}$ from $b$ to $b_\varepsilon^\star$, and applying [Che+20, Lemma 13] we obtain

$$F_\varepsilon(b) - F_\varepsilon(b_\varepsilon^\star) \leq \sqrt{2F_\varepsilon(b)\,W_2(b, b_\varepsilon^\star)\int_0^1 \|\nabla F_\varepsilon(b)\|_{L^2(b_s)}\,\mathrm{d}s}\,.$$

This can be simplified via

$$F_\varepsilon(b) = \int W_{2,\varepsilon}(b, \mu)\,\mathrm{d}P(\mu) \geq W_{2,\varepsilon}(b, b_\varepsilon^\star) - \int W_{2,\varepsilon}(b_\varepsilon^\star, \mu)\,\mathrm{d}P(\mu)\,,$$

and since $b_\varepsilon^\star$ is assumed to be a minimizer of $F_\varepsilon$, then

$$W_2(b, b_\varepsilon^\star) \leq 2F_\varepsilon(b)\,.$$

Hence,

$$F_\varepsilon(b) - F_\varepsilon(b_\varepsilon^\star) \leq 2F_\varepsilon(b)\sqrt{\int_0^1 \|\nabla F_\varepsilon(b)\|_{L^2(b_s)}\,\mathrm{d}s}\,.$$

Next, we specialize the result to the Bures-Wasserstein space. In this case, using the assumptions on $P$ and $\Sigma$, we can argue as in [Che+20, Theorem 19] that $\|\nabla F_\varepsilon(\Sigma)\|_{\Sigma_s}^2 \leq \kappa\,\|\nabla F_\varepsilon(\Sigma)\|_\Sigma^2$, which completes the proof. $\qquad\square$

In order to proceed with the analysis, we must study the dynamics of the smoothed Riemannian GD algorithm to see if we can satisfy the hypotheses of Lemma 12. To study these dynamics, it is helpful to again adopt the notation and calculus of general Wasserstein space. The Wasserstein gradient of $F_\varepsilon$ is

$$\nabla F_\varepsilon(b) = -\int \frac{T_{b\to\mu} - \mathrm{id}}{W_{2,\varepsilon}(b, \mu)}\,\mathrm{d}P(\mu)\,,$$

and one step of the Riemannian GD iteration with step size $\varepsilon$ (which is motivated by Lemma 11) is

$$b^+ := \exp\bigl(-\varepsilon\nabla F_\varepsilon(b)\bigr) = \Big[\mathrm{id} + \varepsilon\int \frac{T_{b\to\mu} - \mathrm{id}}{W_{2,\varepsilon}(b, \mu)}\,\mathrm{d}P(\mu)\Big]_\#\,b\,.$$

We will rewrite this in the following way. Define the weight

$$\rho(\mu) := \frac{W_{2,\varepsilon}(b,\mu)^{-1}}{\int W_{2,\varepsilon}(b,\cdot)^{-1}\,\mathrm{d}P}\,.$$

Then,

$$\mathrm{id} + \varepsilon \int \frac{T_{b\to\mu} - \mathrm{id}}{W_{2,\varepsilon}(b,\mu)}\,\mathrm{d}P(\mu)$$

$$= \Big(1 - \varepsilon \int W_{2,\varepsilon}(b,\cdot)^{-1}\,\mathrm{d}P\Big)\,\mathrm{id} + \Big(\varepsilon \int W_{2,\varepsilon}(b,\cdot)^{-1}\mathrm{d}P\Big) \int T_{b\to\mu}\,\rho(\mu)\,\mathrm{d}P(\mu)\,.$$

Since $\int W_{2,\varepsilon}(b,\cdot)^{-1}\,\mathrm{d}P \le 1/\varepsilon$, this is a convex combination of two terms. Let us call the weights $1-\lambda$ and $\lambda$ respectively. If we define the probability measure $\tilde{P} := (1-\lambda)\delta_b + \lambda\rho P$, then this can also be written as

$$b^+ = \Big(\int T_{b\to\mu}\,\mathrm{d}\tilde{P}(\mu)\Big)_{\#}b\,.$$

This expression proves the following fact (see also Appendix A.2).

**Lemma 13.** *The next iterate $b^+$ of smoothed Riemannian GD starting at $b$ (with step size $\varepsilon$) is a generalized barycenter of the distribution $\tilde{P}$ with base $b$.*

Combined with our geodesic convexity result (Theorem 1), we can now conclude the following important facts about the dynamics of smoothed GD.

**Corollary 1.** *Assume that all of the covariance matrices in the support of $P$ have eigenvalues which lie in the range $[\lambda_{\min}, \lambda_{\max}]$, and that we initialize Algorithm 4 at an element of $\mathrm{supp}\,P$. Then, all of the iterates of Algorithm 4 satisfy the same eigenvalue bounds.*

We can now prove Theorem 5.

*Proof of Theorem 5.* As discussed after Lemma 10, it suffices to find the number of iterations until we find an $\varepsilon$-approximate minimizer of $F_\varepsilon$. Combining the smoothness (Lemma 11) and gradient domination (Lemma 12), as well as Corollary 1, we obtain

$$F_\varepsilon(\Sigma_{t+1}) - F_\varepsilon(\Sigma_t) \le -\frac{\varepsilon}{2}\,\|\nabla F_\varepsilon(\Sigma_t)\|_{\Sigma_t}^2 \le -\frac{\varepsilon}{32\kappa\,F_\varepsilon(\Sigma_0)^4}\,\{F_\varepsilon(b_t) - F_\varepsilon(b^\star)\}^4\,.$$

Let us write

$$\delta_t := F_\varepsilon(\Sigma_t) - F_\varepsilon(\Sigma_\varepsilon^\star)\,, \qquad \zeta := \frac{\varepsilon}{32\kappa\,F_\varepsilon(\Sigma_0)^4}\,.$$

Then, we can rewrite the recursion as

$$\delta_{t+1} \le \delta_t - \zeta\delta_t^4\,.$$

We solve the recursion via induction; we claim that

$$\delta_t \le \frac{1}{\zeta^{1/3}\,(1+t)^{1/3}}\,.$$

This holds when $t = 0$ because $\zeta^{-1/3} \ge \delta_0$. Assuming that the inequality holds at some iteration $t$, we proceed to verify this for iteration $t+1$. If $\delta_t \le 1/[\zeta^{1/3}\,(2+t)^{1/3}]$, then this is immediate because $\delta_{t+1} \le \delta_t$. Otherwise, $\delta_t \ge 1/[\zeta^{1/3}\,(2+t)^{1/3}]$, and we obtain

$$1 - \zeta\delta_t^3 \le 1 - \frac{1}{2+t} = \frac{1+t}{2+t} \le \Big(\frac{1+t}{2+t}\Big)^{1/3}$$

and hence

$$\delta_{t+1} \le \delta_t\,(1 - \zeta\delta_t^3) \le \frac{1}{\zeta^{1/3}\,(1+t)^{1/3}}\,\Big(\frac{1+t}{2+t}\Big)^{1/3} = \frac{1}{\zeta^{1/3}\,(2+t)^{1/3}}\,.$$

The theorem now follows. □

Finally, we finish the last part of Proposition 2.

*Proof of Proposition 2 (Continued).* Let $(\varepsilon_k)_{k \in \mathbb{N}}$ be a sequence of positive numbers tending to zero. The guarantee for smoothed GD in Theorem 5, along with the control over the iterates in Corollary 1, allows us to assert that for each $k \in \mathbb{N}$, there is a point $\Sigma_k \in \mathbb{S}_{++}^d$ with eigenvalues in $[\lambda_{\min}, \lambda_{\max}]$ (the output of smoothed GD) with suboptimality gap $F(\Sigma_k) - \inf F \le \varepsilon_k$. By compactness, we can extract a convergent subsequence of $(\Sigma_k)_{k \in \mathbb{N}}$, which must therefore converge to a minimizer of $F$ (since $F$ is continuous). The limit point must have eigenvalues in the range $[\lambda_{\min}, \lambda_{\max}]$, which completes the proof. $\qquad\square$

### E.2 Reduction for non-zero means

In this section, we suppose that $P$ is supported on non-degenerate, not necessarily centered Gaussians, whose covariance matrices have eigenvalues in the range $[\lambda_{\min}, \lambda_{\max}]$. We begin with the observation that if $b_{\mathrm{median}}^\star$ denotes a Gaussian minimizer of the median functional for $P$, then the mean of $b_{\mathrm{median}}^\star$ is not necessarily the Euclidean geometric median of the means of distributions in $\mathrm{supp}\, P$. To see this, consider the case when the Gaussians are one-dimensional. Then, if we identify each Gaussian $\mu \in \mathrm{supp}\, P$ with its mean and *standard deviation* (the square root of the variance) $(m_\mu, \sigma_\mu)$, then the $W_2$ distance between Gaussians is isometric to the standard Euclidean metric on the pairs $(m, \sigma)$ in $\mathbb{R}^2$ (see Facts 4 and 5 in Appendix A.5). Therefore, the Wasserstein geometric median of $P$ is equivalent to the Euclidean geometric median of the pairs $(m, \sigma)$, and the statement whose validity is being investigated is tantamount to asking: is the first coordinate of the Euclidean geometric median in $\mathbb{R}^2$ equal to the median of the first coordinates? This statement is manifestly false.

Next, we describe the reduction. Let $(m, \Sigma)$, $(m', \Sigma')$ denote two pairs of means and covariance matrices in the support of $P$. Then,

$$W_2^2\big((m, \Sigma), (m', \Sigma')\big) = \|m - m'\|^2 + W_2^2(\Sigma, \Sigma')\,,$$

where the LHS denotes the squared Wasserstein distance between Gaussians with parameters $(m, \Sigma)$ and $(m, \Sigma')$ respectively. The idea behind the reduction is that since the Wasserstein metric on diagonal matrices is the same as the Euclidean metric between the *square roots* of the matrices (Fact 4 in Appendix A.5), we can embed the mean vectors as diagonal matrices, and take the direct sum of these diagonal matrices with the covariance matrices to form augmented matrices; then, we can apply the geometric median algorithm (Algorithm 4) to the augmented matrices. In this reduction, however, we must take care that when we embed the mean vectors, we embed them into *positive definite* diagonal matrices.

Hence, define the augmented matrices

$$\mathbf{\Sigma} := \begin{bmatrix} \mathrm{diag}((m + C)^2) & \\ & \Sigma \end{bmatrix}, \qquad \mathbf{\Sigma}' := \begin{bmatrix} \mathrm{diag}((m' + C)^2) & \\ & \Sigma' \end{bmatrix},$$

where the constant $C \ge \sqrt{\lambda_{\min}} + \max\{\|m\|_\infty, \|m'\|_\infty\}$ is chosen to ensure that $\mathbf{\Sigma}, \mathbf{\Sigma}' \succeq \lambda_{\min} I_d$ (and that $m + C, m' + C \ge 0$). The Wasserstein distance between the augmented matrices is

$$W_2^2(\mathbf{\Sigma}, \mathbf{\Sigma}') = \|(m + C) - (m' + C)\|_{\mathrm{F}}^2 + W_2^2(\Sigma, \Sigma') = \|m - m'\|^2 + W_2^2(\Sigma, \Sigma')$$
$$= W_2^2\big((m, \Sigma), (m', \Sigma')\big)\,.$$

Hence, after preprocessing the mean vectors and covariance matrices to form these augmented matrices, we may apply Algorithm 4 to the augmented matrices in a black box manner. It is easy to check that the set of such diagonal block matrices (where the upper block is itself diagonal) is convex under generalized geodesics. Hence, as long as the Algorithm 4 is initialized at such a matrix every iterate will remain in that form, and therefore the iterates will, when transformed back through the augmentation operation described above, indeed approach optimality for the original median problem. Note that the new value of the condition number will be

$$\boldsymbol{\kappa} = \max\Big\{\kappa, \big(1 + \frac{2B}{\sqrt{\lambda_{\min}}}\big)^2\Big\}, \qquad B := \sup_{\mu \in \mathrm{supp}\, P} \|m_\mu\|_\infty\,.$$

The convergence guarantee of Theorem 5 applies, with $\boldsymbol{\kappa}$ replacing $\kappa$.

Of course, it is likely that analyzing smoothed Riemannian GD directly for the non-centered case could produce sharper results (in particular, with a dependence on $\kappa$ rather than $\boldsymbol{\kappa}$), but this simple approach already gives dimension-free convergence rates for the Bures-Wasserstein geometric median.

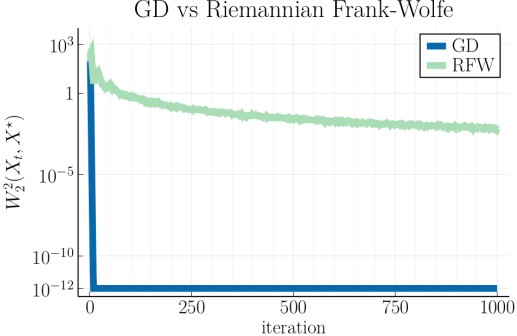

Figure 7: Riemannian GD vs Riemannian Frank-Wolfe, for computing Bures-Wasserstein barycenters.

# F    Further experiments and details

**Reproducibility details.**    Input generation details for Figures 1, 2, 5, and 6 are provided in the main text. For Figures 3 and 4, recall that we generated matrices from a distribution whose barycenter is known to be the identity. By [ZP19, Theorem 2], if the mean of the distribution $(\log_{I_d})_\# P$ is 0, then $I_d$ is the barycenter of $P$. In particular, if $Q$ is a mean zero distribution supported on symmetric matrices that lie in the domain of the exponential map, then $P = (\exp_{I_d})_\# Q$ has $I_d$ as its barycenter. In our experiments, we defined $Q$ to be the law of a random matrix with Haar eigenbasis and uniform eigenvalues from the interval $[-(1-\delta), 1-\delta]$ for a parameter $\delta \in (0,1)$. At the identity, the exponential map takes the simple form $\exp_{I_d} S = (I_d + S)^2$ and we see that $P$ is then supported on covariance matrices with spectrum in $[\delta^2, (2-\delta)^2]$. Both figures were generated with $\delta = 0.1$. All experiments were performed using Julia 1.5.1 on a desktop computer running Ubuntu 18.04 with an Intel i7-10700 CPU.

**Riemannian Frank-Wolfe.**    Here we provide an empirical comparison with the Riemannian Frank-Wolfe algorithm for computing Bures-Wasserstein barycenters, as described in [WS17, Algorithm 3]. Figure 7 demonstrates the superior practical performance of Riemannian GD, the algorithm studied in this paper. In this experiment, the input is as in Figure 2, and $X^\star$ denotes the best iterate.

**Further empirical comparisons.**    Here we further investigate the comparison of Riemannian and Euclidean GD done in Figure 2 by demonstrating qualitatively similar results for a variety of synthetic datasets. For each dataset, the measure $P$ is the empirical measure of $n$ matrices of dimension $d \times d$ that are drawn randomly as follows.

1. Haar eigenbasis and linearly spaced eigenvalues in $[\alpha, \beta]$.

2. Haar eigenbasis and i.i.d. $\mathrm{Unif}[\alpha, \beta]$ eigenvalues.

3. First split the matrices into 3 groups. Each matrix has Haar eigenbasis and i.i.d. $\mathrm{Unif}[\alpha, \beta]$ eigenvalues where $[\alpha, \beta] = 10^i \times [1, \kappa]$ for $i \in \{-2, 0, 2\}$ depending on its group.

4. Same as method 2 above, except all matrices have the same eigenbasis. (Note that GD converges in 1 step here since the matrices commute.)

5. Haar eigenbasis and eigenvalues uniform on a set of size $m \leq d$, whose elements are i.i.d. $\mathrm{Unif}[\alpha, \beta]$.

6. Same as method 5 above, except all matrices use the same eigenvalues.

7. Mix of all methods above.

Figures 8a and 8b compare Euclidean and Riemannian GD on the barycenter problem as in Figure 2, but now with these 7 different input families. We average well-conditioned matrices in Figure 8a, and ill-conditioned matrices in Figure 8b. The plots are generated using $n = d = 50$ and $m = d/4$. For Method 7, the 50 matrices are divided into 6 groups of roughly equal size. The $y$-axis measures the $W_2^2$ distance to the best iterate; and the $x$-axis measures time in seconds.

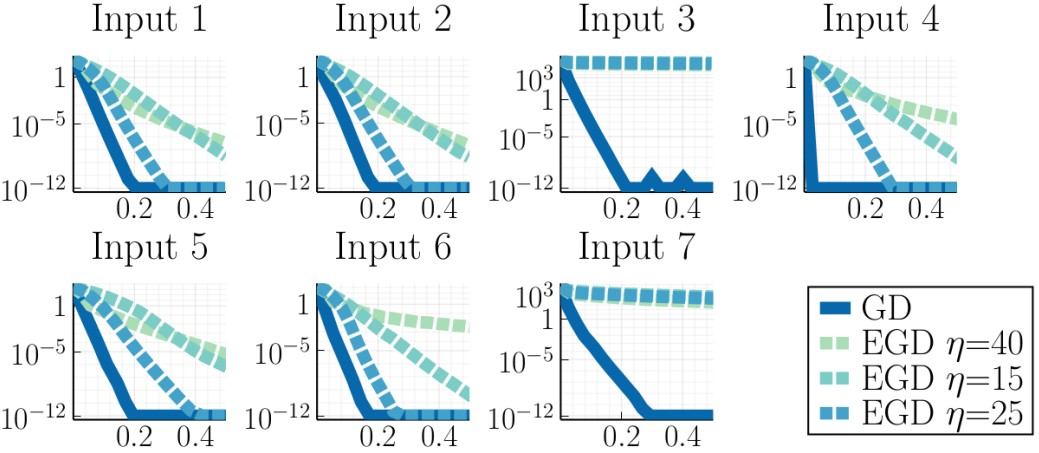

(a) Here, the matrices are poorly conditioned, namely $[\alpha, \beta] = [0.03, 30]$ whereby $\kappa = 1000$.

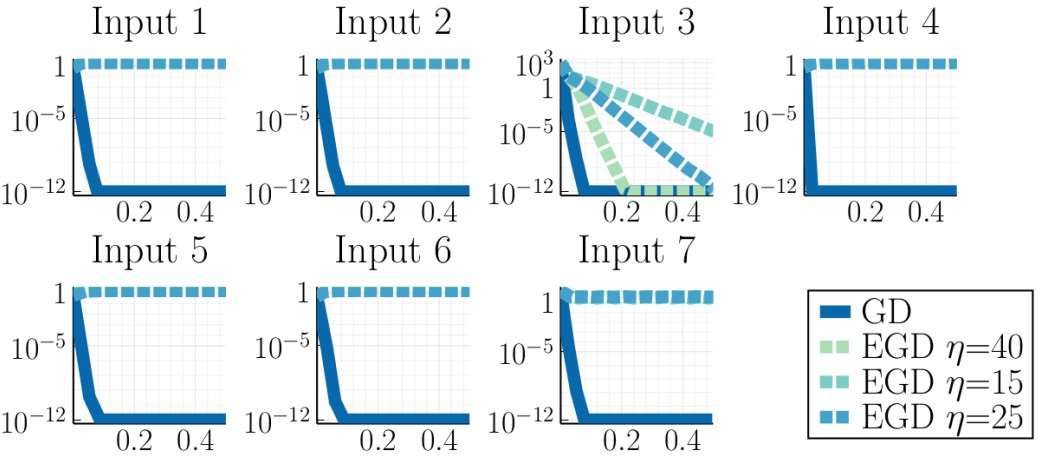

(b) Here, the matrices are well-conditioned, namely $[\alpha, \beta] = [1, 2]$ whereby $\kappa = 2$.

Figure 8: Comparison of high-precision barycenter algorithms for various types of synthetic data.

In these figures we had to hand-tune the stepsize for Euclidean GD since the stepsize indicated by Theorem 9 performs quite poorly. We used the same range of stepsizes ($\eta \in \{15, 25, 40\}$) in all plots to demonstrate that the performance of Euclidean GD is quite sensitive to its stepsize. In contrast, GD performs well on all inputs with its (untuned) stepsize of 1.