# OpenReview forum: "Averaging on the Bures-Wasserstein manifold: dimension-free convergence of gradient descent"
_NeurIPS.cc/2021/Conference — NeurIPS 2021 Spotlight_

### Official Review · Reviewer_vGkj · 2021-07-09

**Rating:** 8
**Confidence:** 3

**Summary:**


The subject of this paper is Riemannian (stochastic) gradient descent for Bures-Wasserstein barycenters of Gaussian measures. The authors first improve recent results by S. Chewi et al (2020) by removing the exponential dependence of the convergence rates w.r.t. the dimension d. This improvement is obtained by showing the geodesic convexity of $-\sqrt{\lambda_{\min}}$ and $\sqrt{\lambda_\max}$, which yields tighter bounds on the eigenvalues of the iterates of Bures-(S)GD than that of S. Chewi et al (2020). Next, the authors extend Bures-(S)GD to a KL-regularized barycenter problem that encodes an isotropy prior on the barycenter, and to a median version of the barycenter problem (where the sum of un-squared distances is minimized). Finally, in numerical experiments they compare Bures-Riemannian (S)GD to Euclidean (S)GD on the original barycenter problem, and investigate the effect of smoothness parameters in the regularized and median problems.


**Limitations And Societal Impact:**

My only comment on that subject is that the fact that the algorithm for geometric medians computes an approximate solution that depends on the smoothness parameter $\varepsilon$ could be mentioned earlier than in section 5. However, based on Figure 5 it seems that numerical precision is reached quite fast when taking $\varepsilon$ to $0$.

**Main Review:**

I have greatly enjoyed reading this paper, which is well-written and makes important contributions, even though it relies on techniques developed by S. Chewi et al (2020). Here is a list of points I particularly appreciated:

- Removing the exponential dependence in $d$ is a *tour de force* that closes the gap with the sample complexity bounds for Bures-Wasserstein distance estimation, which are also dimension-free (see e.g. R. Flamary et al. (2019))

- The paper is very clearly-written, and mostly self-contained. I particularly appreciated the fact that all required geometric notions are recalled in appendix A, which does a great job at introducing them in an intuitive way.

- The extensions to the regularized barycenter and geometric median are interesting, in that they show how the tools developed for the Riemannian descent on the BW barycenter problem can be applied to problems that seem less amenable to them at first sight.

- Thm. 1 is of interest by itself, and could probably be used to optimize spectral functions of PSD matrices in the Bures geometry.


A few comments:

- While I appreciated the fact that they provide additional examples of how to apply proof techniques based on (generalized) geodesic convexity, I am not sure how significant the regularized barycenter and geometric problems are.

- l.238: « Prior work focuses on a slightly different entropic penalty, the differential entropy  $\int b \ln b$. ». There are some works that study this problem with a broader class of penalties, see e.g. the work of J. Bigot et al. (2019) which seems quite relevant.


References:

Flamary, R., Lounici, K., & Ferrari, A. (2019). Concentration bounds for linear Monge mapping estimation and optimal transport domain adaptation. arXiv preprint arXiv:1905.10155.

Bigot, J., Cazelles, E., & Papadakis, N. (2019). Penalization of barycenters in the Wasserstein space. SIAM Journal on Mathematical Analysis, 51(3), 2261-2285.

**Time Spent Reviewing:**

5

---

> ### Author Response · Authors · 2021-08-11
> **Response**
>
> Thank you for the very kind comments. We are glad that you enjoyed reading our paper.
>
> In our camera-ready version, we will add comments about all these good points. Especially the relevant reference to Bigot et al., and the fact that our dimension-free algorithmic result complements the dimension-free statistical result of Flamary et al.
>
> Re: question about the smoothness parameter. Thank you for pointing out the possible confusion arising from the smoothing of the median functional. Theorem 5 shows that our solution (optimized for F_ε) is an approximate minimizer of the unregularized functional F. Exactly as you point out, our empirics give promising demonstrations that our algorithm can perform significantly better in practice than our theoretical guarantees, both in that few iterations suffice for convergence and also that moderate regularization suffices for high-precision approximations. We’ll clarify this in the camera-ready.

---

> > ### Comment · Reviewer_vGkj · 2021-08-20
> > **Post-rebuttal comment**
> >
> > Thanks to the authors for answering my comments. I have read all reviews and their responses. I think this is a great theoretical paper, and maintain my score.

---

### Official Review · Reviewer_t1br · 2021-07-14

**Rating:** 8
**Confidence:** 5

**Summary:**

This work is with the recent active wave of work in numerical optimal transport to take ML and statistical estimation problems. More precisely, the authors study first-order optimization algorithms on Bures-Wasserstein manifold for computing the barycenter of Gaussian measures with respect to the 2-Wassertstein distance, the  regularized barycenter and geometric median.


**Limitations And Societal Impact:**

None.

**Main Review:**

The main contribution of the paper is to show new geodesic convexity results hence allowing to obtain a dimension-free convergence rate of Riemannian gradient descent. The techniques also enable to go beyond barycenters by tackling the problem the entropically-regularized barycenter and geometric median, hence providing first convergence guarantees for these problems. Although linear convergence of gradient descent on the Bures-Waasserstein manifold was established recently, the rate had exponential dependence on the dimension. Therefore, as far as I can say, the results of this paper are new and of interest to the literature. The paper is also fairly well-written. Having said this, I still have some comments:

1) I found that the paper lacks discussion to prior work for each stated result (though a brief discussion is gathered in the introduction).
2) P3, L106-107: the reference given there is not fair. Rather cite the original paper of Łojasiewicz, and those of J. Bolte and his co-authors who have the actual pioneers of this in optimization.
3) P4, L151-153: please void self-citation for such a classical subject on which there are many standard monographs (including diCarmo).
4) P4, L155: sentence in parentheses. You say a bit later that it is in [Che+20, §4]. Note also that for the regularized W_2, it is also in [Jan+20, Lemma 1].
5) P5, Theorem 2: the notation of the iteration counter is t in the algorithms and n here. Unify notations.
6) P5, Theorem 2: can you comment on optimality of this rate ? We know the optimal rate in the Euclidean (convex) case.
7) P5, Theorem 2: the statement "appropriate step-sizes" (ans also the K_t's) has to be made precise here (not just in the appendix). It is well-known that such a choie crucial to get the 1/n rate.
BTW, can the dependence in \kappa^3 be improved again ?
Is there any hope to have a variance reduced version for the case of finitely many distributions (i.e. finite sum) ?
8) Figure 2: the difference between ESGD and SGD is quite sensitive to the constant in the stepsize. Can you comment on this ?
9) P6, L208: n is now used for the number of Gaussians (see comment 5) above).
10) Figure 3 and dimension-independence: an alternative way would be rather to run several examples of the experiment on Figure 1 for increasing dimension, estimate the rate from a semilog fit, and plot the estimated rate as a function of d.
11) Proposition 1: a similar result is in Theorem 2 of [Jan+20] but with a different metric (debiased version of the entropically regularized 2-W). Discuss please.
12) Theorem 5: why this form of smoothing ? There are other ways, probably with better guarantees, or even no smoothing with subgradient descent. the iteration complexity in \varepsilon^{-4} is rather slow (and play be pessimistic). Any clue on optimality of this bound or if it is unavoidable ?


**Time Spent Reviewing:**

3h.

---

> ### Author Response · Authors · 2021-08-11
> **Response**
>
> Thank you for the very kind comments.
>
> - 1, 2, 3, 4, 5, expository part of 7, 9, experiments in 10. Great suggestions. We will make all these changes in the camera-ready version; this will improve the exposition.
>
> - 6, 7. Re: optimality of our rates, this is a very interesting direction for future work. The purpose of our paper was to improve the dimension dependence from exponential to dimension-free. While we also improved the dependence on the condition number, it is unclear whether this is optimal.
>
> - 8: Our theorems give step sizes for SGD and ESGD with provable worst-case guarantees. As Figure 2 shows, tuning the stepsizes further can help in practice. We note that SGD seemed competitive on a range of problems with η_t = 1/t. As an aside, a GD requires no step size tuning at all.
>
> - 11: Thm. 2 of [Jan+20] expresses the debiased entropic W2 barycenter as the solution to a nonlinear equation (“gradient = 0”). While the entropic W2 barycenter can be expressed similarly, such a representation does not appear to be useful for the purpose of computation.
>
> - 12: We chose our form of smoothing mainly out of convenience for the analysis in order to illustrate that this simple approach already yields dimension-independent rates for the Wasserstein geometric median. Since the unsmoothed geometric median functional is non-convex, we are not aware of how to obtain guarantees for subgradient descent. Investigating other methods of smoothing is an interesting direction for future work.

---

> > ### Comment · Reviewer_t1br · 2021-08-23
> > **Overall opinion.**
> >
> > After reading the author's rebuttal, I am completely satisfied by the answers given and pretty sure the authors will make the necessary job to implement them. Therefore, I maintain my score.

---

### Official Review · Reviewer_Bkse · 2021-07-15

**Rating:** 3
**Confidence:** 1

**Summary:**

The paper address the problem of computing the barycenter of a gaussian distributions using optimal transport metric. The paper propose two algorithm (section 3.1, section 4.1) to compute thw Entropically regularized barycenters and the Bures-Wasserstain barycenters and it studies its convergence guarantees.

**Limitations And Societal Impact:**

No, unfortunately the discussion is missing.

**Main Review:**

Even if the paper deals with an interesting and important topic, it is not well organized neither clear. For example, the general discussion is completely missing. In this current form, the paper can not be published. As a general comment,  I think it is more suitable for a journal, where there is more room for discussion of the literature and additional details.

Some comments for the possible reorganization of the paper:
1) Section 1 and Section 2 should be reorganized, separating the state of the art from the rest
2) Results in section 3,4,5 should be well listed at the beginning, otherwise it is not clear the whole paper structure
3) Discussion is a fundamental part of the paper, beacuse it allows the reader to understand the work and value your discovers
4) The simulated data should be better explained, not left on the side of the figures as captions.



**Time Spent Reviewing:**

1

---

> ### Author Response · Authors · 2021-08-11
> **Response**
>
> Thank you for the expository suggestions. We will make all of these changes in the camera-ready.
>
> We are sorry that you found the paper unclear -- we are happy to answer any questions about the content if you have any.

---

### Official Review · Reviewer_A7RV · 2021-07-16

**Rating:** 7
**Confidence:** 4

**Summary:**

The authors study Riemannian gradient descent on the Bures-Wasserstein manifold and provide non-asymptotic convergence guarantees for three optimization tasks: The computation of Wasserstein barycenters, entropy-regularized Wasserstein barycenters, and geometric medians.

**Limitations And Societal Impact:**

yes

**Main Review:**

After rebuttal: Thanks to the authors for their comments. I'm keeping my score of acceptance.

-----------------------------------------

*Assessment*

Non-asymptotic convergence guarantees for regularized barycenters and geometric medians are to my knowledge novel and should be of great interest to the community. The theoretical results are clearly formulated and seem correct. The paper is very well written.

*Additional Comments*

- Related work: Wasserstein barycenters on the Bures manifold lie in a compact, geodesically convex set and can therefore be treated as a constrained problem. One can then get non-asymptotic convergence guarantees with constrained Riemannian methods, such as Frank-Wolfe. The PL inequality should ensure linear convergence for such methods too. See, e.g., [*].
- I did not see any experimental results for a practical application. The current numerical evaluation on synthetic data is (to me) sufficient for acceptance, but an experiment illustrating the results in an application could strengthen the paper.

[*] Weber, Sra: Projection-free nonconvex stochastic optimization on Riemannian manifolds


**Time Spent Reviewing:**

---

> ### Author Response · Authors · 2021-08-11
> **Response**
>
> Thank you for the very kind comments.
>
> Re: Riemannian Frank-Wolfe, it is not clear to us that dimension-independent rates can be obtained via RFW, although your suggestion is an intriguing direction for future work. Following much of the literature, we focus on Riemannian GD due to its strong empirical performance. (Preliminary numerical experiments indicate that RGD is at least an order of magnitude faster than RFW; we will add a comparison to the appendix in the camera-ready version.)
>
> Re: practical experiments, good point. The main purpose of this paper is to provide a theoretical analysis of RGD, and the intention of our experiments is to numerically validate our theoretical results. As such, developing details of applications is an important direction, but out of the scope of this paper.

---

### Decision · Program_Chairs · 2021-09-27

**Decision:**

Accept (Spotlight)

**Comment:**

The paper proves dimension-free convergence rates for the gradient descent algorithm for the Bures-Wasserstein Barycenter computation. It is a significantly novel result that provides a theoretical basis to something that is practically observed and as such should be of interest to many.